# Loss of flavin adenine dinucleotide (FAD) impairs sperm function and male reproductive advantage in *C. elegans*

Chia-An Yen[1,2], Dana L Ruter[1,2], Christian D Turner[1,2], Shanshan Pang[3], Sean P Curran[1,2,4]*

[1]Leonard Davis School of Gerontology, University of Southern California, Los Angeles, United States; [2]Department of Molecular and Computation Biology, Dornsife College of Letters, Arts, and Sciences, University of Southern California, Los Angeles, United States; [3]School of Life Sciences, Chongqing University, Chongqing, China; [4]Norris Comprehensive Cancer Center, Keck School of Medicine, University of Southern California, Los Angeles, United States

**Abstract** Exposure to environmental stress is clinically established to influence male reproductive health, but the impact of normal cellular metabolism on sperm quality is less well-defined. Here we show that impaired mitochondrial proline catabolism, reduces energy-storing flavin adenine dinucleotide (FAD) levels, alters mitochondrial dynamics toward fusion, and leads to age-related loss of sperm quality (size and activity), which diminishes competitive fitness of the animal. Loss of the 1-pyrroline-5-carboxylate dehydrogenase enzyme *alh-6* that catalyzes the second step in mitochondrial proline catabolism leads to premature male reproductive senescence. Reducing the expression of the proline catabolism enzyme *alh-6* or FAD biosynthesis pathway genes in the germline is sufficient to recapitulate the sperm-related phenotypes observed in *alh-6* loss-of-function mutants. These sperm-specific defects are suppressed by feeding diets that restore FAD levels. Our results define a cell autonomous role for mitochondrial proline catabolism and FAD homeostasis on sperm function and specify strategies to pharmacologically reverse these defects.

*For correspondence:
spcurran@usc.edu

Competing interests: The authors declare that no competing interests exist.

## Introduction

As individuals wait longer to have families, reproductive senescence has become an increasingly prudent topic (*Mills et al., 2011*; *Lemaître and Gaillard, 2017*). Decline in oocyte quality is well-documented with age and can result in fertility issues when older couples try to conceive (*Baird et al., 2005*). Furthermore, pregnancies at an older age pose risks for higher incidences of birth defects and miscarriages. In humans, female reproduction ceases at an average age of 41–60, with the onset of menopause (*Treloar, 1981*). The *Caenorhabditis elegans* 'wild type' is hermaphroditic and self-fertilizing; however, they are capable of making and maintaining Mendelian ratios of male (sperm-only) animals in their populations. Like humans, *C. elegans* experience a decline in fecundity with age by halting oocyte production at roughly one-third of their lifespan (*Kadandale and Singson, 2004*). In addition, regulators of reproductive aging, such as insulin/IGF-1 and *sma*-2/TGF-β signaling, are conserved regulators of reproductive aging from worms to humans (*Luo et al., 2010*). While the majority of studies in reproductive senescence have focused on maternal effects, male factors contribute to a large portion of fertility complications with increasing evidence of an inverse relationship between paternal age and sperm health (*Lemaître and Gaillard, 2017*). In fact, studies in mammals have shown an age-related decline in sperm quality with increased incidences of DNA damage, reduced motility, abnormal morphology, and decreased semen volume (*Cocuzza et al., 2008*; *Kidd et al., 2001*; *Ozkosem et al., 2015*).

Flavin adenine dinucleotide (FAD) is an important cofactor that participates in enzymatic redox reactions that are used in cellular metabolism and homeostasis. FAD is synthesized from riboflavin by the concerted actions of FAD synthetase and riboflavin kinase. Like humans, *C. elegans* cannot synthesize riboflavin, and therefore requires dietary intake (*Braeckman, 2009*). Disruption of flavin homeostasis in humans and animal models has been associated with several diseases, including: cardiovascular diseases, cancer, anemia, abnormal fetal development, and neuromuscular and neurological disorders (*Barile et al., 2013*); however, the link between FAD homeostasis and fertility is undefined.

We demonstrate that, although reproductive senescence is generally studied only from the female viewpoint, age-specific female reproductive success strongly depends on male–female interactions. Thus, a reduction in male fertilization efficiency with increasing age has detrimental consequences for female fitness. Lastly, we call for investigations of the role of environmental conditions on reproductive senescence, which could provide salient insights into the underlying sex-specific mechanisms of reproductive success.

Several studies have documented fertility defects in *C. elegans* mitochondrial mutants. Mutation in *nuo-1*, a complex I component of the mitochondria respiratory chain, results in reduced brood size caused by impaired germline development (*Grad and Lemire, 2004*). Similarly, *clk-1* mutation affects the timing of egg laying, resulting in reduced brood size (*Jonassen et al., 2002*). Both of these mitochondrial mutations impact fertility, but their role(s) in spermatogenesis are unclear. *alh-6*, the *C. elegans* ortholog of human *ALDH4A1*, is a nuclear-encoded mitochondrial enzyme that functions in the second step of the proline metabolism pathway, converting 1-pyrroline-5-carboxylate (P5C) to glutamate (*Adams and Frank, 1980*). We previously revealed that *alh-6(lax105)* loss-of-function mutants display altered mitochondrial structure in the muscle accompanied by increased level of ROS in adult animals (*Pang and Curran, 2014*). Furthermore, mutation in *alh-6* results in the activation of SKN-1/NRF2 (*Pang et al., 2014*), an established regulator of oxidative stress response, likely through the accumulation of toxic P5C disrupting mitochondrial homeostasis (*Pang and Curran, 2014*; *Pang et al., 2014*; *Deuschle et al., 2004*; *Miller et al., 2009*; *Nomura and Takagi, 2004*). Interestingly, SKN-1 was recently shown to respond to accumulation of damaged mitochondria by inducing their biogenesis and degradation through autophagy (*Palikaras et al., 2015*). Here, we identify a genetic pathway that regulates male reproductive decline stemming from the perturbation of mitochondrial proline metabolism leading to redox imbalance, cofactor depletion, and altered mitochondria dynamics; all of which play a role in sperm dysfunction.

## Results

### Mutation in mitochondrial *alh-6* results in diet-independent reduction in fertility

Altered mitochondrial structure and activity have been correlated with sperm dysfunction across different species (*Liau et al., 2007*; *Amaral et al., 2013*; *Ramalho-Santos and Amaral, 2013*; *Nakada et al., 2006*). In addition, proper sperm function requires low levels of ROS (*de Lamirande and Gagnon, 1993*; *Kodama et al., 1996*; *Leclerc et al., 1997*), although a specific role for endogenous mitochondrial derived ROS is undefined. ALH-6/ALDH4A1, is a nuclear-encoded mitochondrial enzyme that functions in the second step of proline catabolism, converting 1-pyrroline-5-carboxylate (P5C) to glutamate (*Figure 1A*). We anticipated that mutation of *alh-6* may affect the germline, based on our previous assessment of the premature aging phenotypes in somatic cells of *alh-6* mutants (*Pang and Curran, 2014*). Using an UV-integrated *alh-6::gfp* strain under its endogenous promoter, we saw that ALH-6 localizes to the mitochondria in the germline of both hermaphrodites and males (*Figure 1—figure supplement 1*). We then assessed progeny output of *alh-6(lax105)* hermaphrodites fed the standard OP50/*E. coli* B strain diet and found a reduction in self-fertility brood size (−12.9%) (*Figure 1B*). Since the somatic phenotypes of *alh-6(lax105)* mutants are known to be diet-dependent (*Pang et al., 2014*; *Pang and Curran, 2014*), we examined self-fertility of animals fed the HT115/*E. coli* K-12 strain diet to determine if the reduced reproductive output is also dependent on the type of bacterial diet ingested. Surprisingly, we found that the self-fertility of *alh-6* animals was markedly reduced (−20.7%), when animals were fed the HT115 diet (*Figure 1C*). *alh-6* mutants have similar timing in their progeny output as compared to wild type animals on both diets

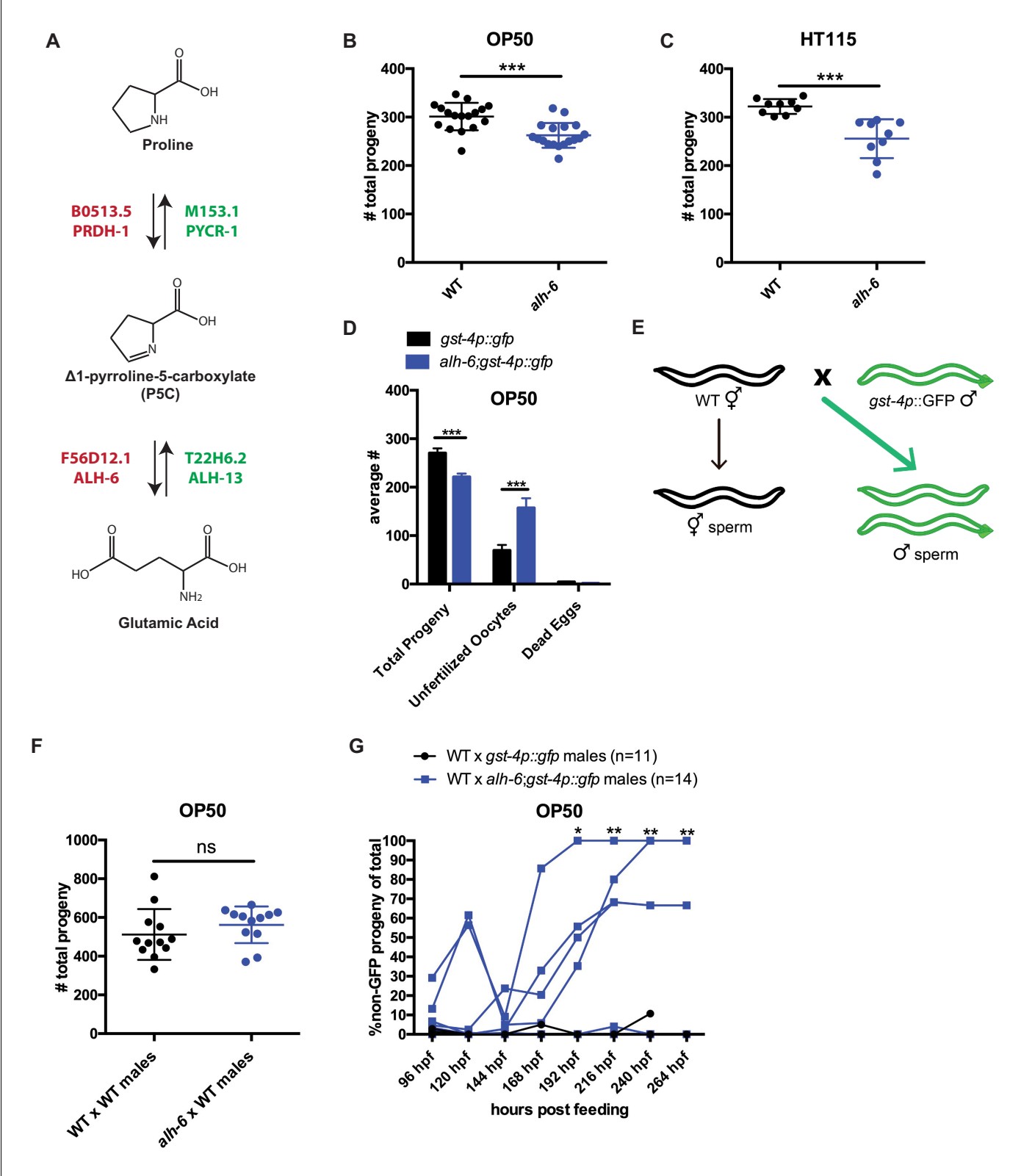

**Figure 1.** *alh-6* fertility defects are sperm-specific. (**A**) Proline catabolism pathway. (**B–C**) *alh-6* hermaphrodites have reduced brood size when fed OP50 (**B**) or HT115 (**C**) diets. (**D**) *alh-6* hermaphrodites lay increased number of unfertilized oocytes, but few dead embryos. (**E**) Mated reproductive assay scheme utilizes males to maximize reproductive output (as in F) and can exploit males harboring GFP to differentiate progeny resulting from self- versus male-sperm (as in G). (**F**) Wild type (WT) and *alh-6* hermaphrodites mated with WT males yield similar number of total progeny. (**G**) WT hermaphrodites

*Figure 1 continued on next page*

*Figure 1 continued*

mated with *alh-6;gst-4p::gfp* males yield more non-GFP progeny (indicating self-fertilization) than hermaphrodites mated with WT males harboring *gst-4p::gfp*. Statistical comparisons by unpaired t-test. *, p<0.05; **, p<0.01; ***, p<0.001; ****, p<0.0001. All studies performed in at least biological triplicate; refer to *Supplementary file 1* for n for each comparison.

The online version of this article includes the following figure supplement(s) for figure 1:

**Figure supplement 1.** Mitochondrial localization of ALH-6 in the germline.
**Figure supplement 2.** *alh-6* hermaphrodite reproductive span is similar to wild type (WT) on different diets.
**Figure supplement 3.** *alh-6* fertility defects are sperm-specific.

(*Figure 1—figure supplement 2*). Since *alh-6* mutants display normal development and reproductive timing, the progeny deficit is not a result of an attenuated reproductive span which reveals the differential impact of *alh-6* loss in the soma (diet-dependent) (*Pang and Curran, 2014*) and the germline (diet-independent).

We noted that *alh-6* mutant hermaphrodite animals laid twice as many unfertilized oocytes as wild type animals over their reproductive-span (*Figure 1D*), suggesting an impairment of sperm function (*McCarter et al., 1999*; *Ward and Miwa, 1978*; *Argon and Ward, 1980*). It is notable that *alh-6* mutant hermaphrodites lay very few, if any, dead eggs (*Figure 1D*), suggesting that the loss of ALH-6 activity is not lethal. To determine whether the reduced brood size of *alh-6* mutants are due to a general loss of germ cells or a specific defect in oocytes or sperm, we examined the mated-fertility of these animals by mating wild type young adult (day 0–1) males to either wildtype or *alh-6* mutant virgin hermaphrodites (in wild type *C. elegans*, male sperm outcompetes hermaphrodite sperm >99% of the time (*Ward and Carrel, 1979*; *LaMunyon and Ward, 1995*; *Figure 1E*). We found that the reduced fertility in *alh-6* mutant hermaphrodites is fully rescued by wild type sperm, which confirmed that oocyte quality is not impaired but rather, *alh-6* hermaphrodite sperm appears to be dysfunctional (*Figure 1F*).

To better assess the quality of *alh-6* mutant sperm, we compared the ability of *alh-6* mutant male sperm to compete against wild type hermaphrodite sperm (*Singson et al., 1999*). In *C. elegans* wild type animals, male sperm are larger and faster than hermaphrodite sperm, which affords a competitive advantage (*LaMunyon and Ward, 1998*). To differentiate between progeny resulting from mating and progeny that arise from hermaphrodite self-fertilization, we made use of male animals harboring a GFP transgene such that any cross-progeny will express GFP while progeny that arise from hermaphrodite self-sperm will not (*Figure 1E*). We found that wild type hermaphrodites when mated to *alh-6* mutant males have significantly more self-sperm-derived progeny as compared to those mated to wild type males (*Figure 1G*). This finding indicates a competition deficit of *alh-6* male sperm resulting in this increased proportion of progeny derived from hermaphrodite sperm, which is uncommon after mating has occurred (*Ward and Carrel, 1979*). *C. elegans* hermaphrodites produce a set amount of sperm exclusively at the L4 developmental stage, before switching exclusively to oogenesis. As such, hermaphrodites eventually deplete their reservoir of sperm (*Hirsh et al., 1976*; *Ward and Carrel, 1979*). To assess whether *alh-6* mutant sperm are generally dysfunctional, we mated older hermaphrodites that had depleted their complement of self-sperm and found that *alh-6* mutant males are able to produce equal numbers of progeny as wild type males when the need for competition with hermaphrodite sperm is abated (*Figure 1—figure supplement 3A*); thus, although *alh-6* mutant sperm are impaired for competition, they remain viable for reproduction. Similarly, older sperm-depleted *alh-6* mutant hermaphrodites produced similar brood sizes when mated to young wild type or *alh-6* mutant males, which further supports a model where sperm, but not oocytes, are defective in *alh-6* mutants (*Figure 1—figure supplement 3B*). Taken together, these data suggest that while *alh-6* mutant male sperm remain competent for fertilization, their competitive advantage is impaired when challenged against hermaphrodite sperm.

## Defects in mitochondrial proline catabolism impact sperm quality

Similar to mammals, the contribution of sperm to fertility in *C. elegans* is dictated by distinct functional qualities, which include: sperm number, size, and motility (*Ward and Miwa, 1978*; *LaMunyon and Ward, 2002*; *LaMunyon and Ward, 1998*; *Singson et al., 1999*). We next sought to define the nature of the sperm competition defect in *alh-6* mutants by measuring sperm number,

size, and motility in *alh-6* mutants compared to wild type animals. One day after the onset of spermatogenesis (at the L4 larval stage of development), *alh-6* adult hermaphrodites have a reduced number of sperm in the spermatheca as compared to wild type (*Figure 2—figure supplement 1A*), which is correlated with the reduced self-fertility observed (*Figure 1B–C*). In contrast, age-matched *alh-6* mutant virgin males have similar numbers of spermatids as WT virgin males, suggesting that they have a similar rate of production (*Figure 2A*). We next examined sperm size in day one adult males and discovered that *alh-6* mutant spermatids are significantly smaller as compared to wild

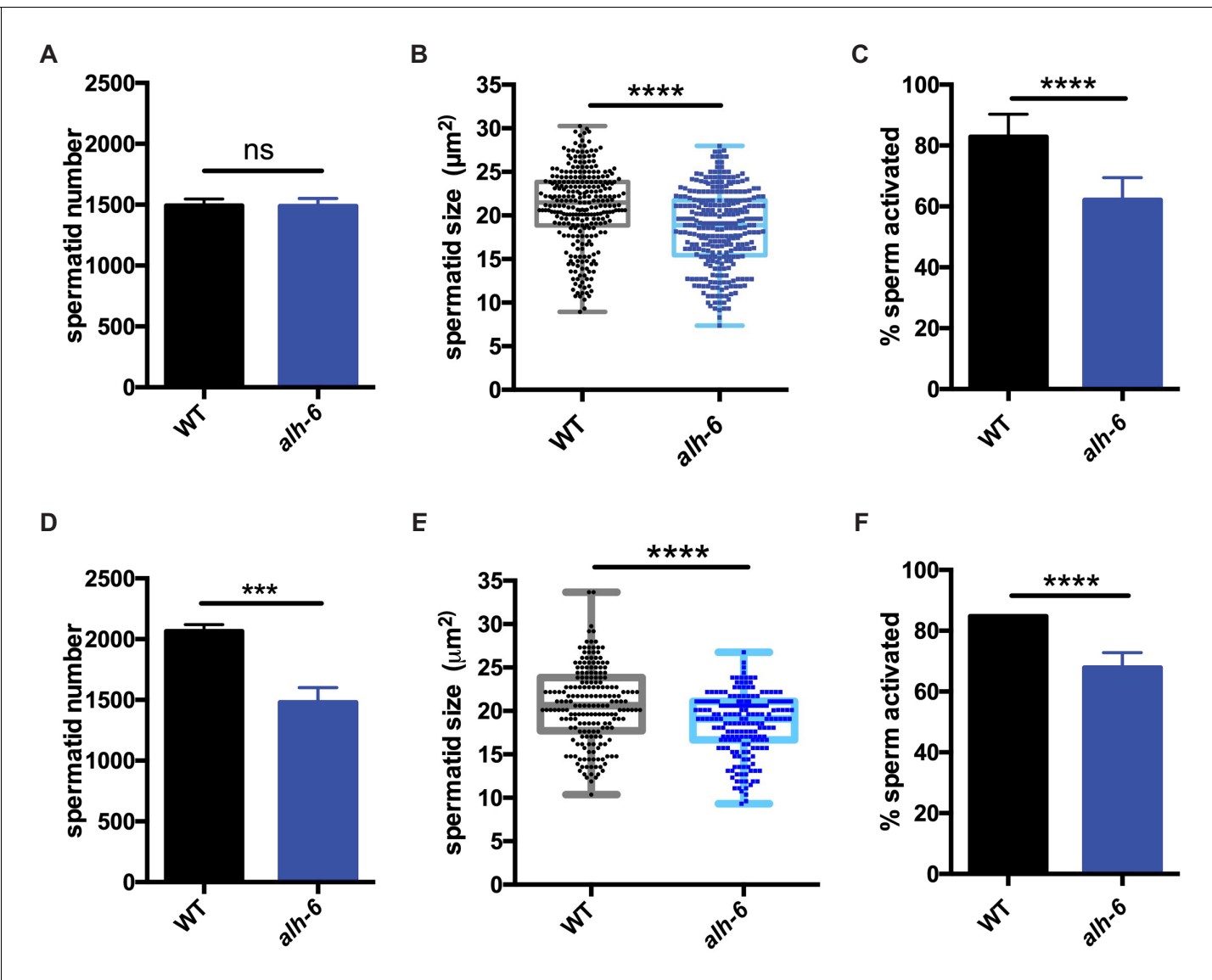

**Figure 2.** *alh-6* males have sperm defects on both OP50 and HT115 diets. (A–C) sperm phenotypes on OP50 diet. (A) Sperm quantity is similar between wild type (WT) and *alh-6* mutant day one adult males. (B) Spermatid size is reduced in *alh-6* mutant day one adult males as compared to age matched WT males. (C) Sperm activation is impaired in *alh-6* mutant day one adult males relative to age-matched WT males. (D–F) sperm phenotypes on HT115 diet. (D) Sperm quantity is reduced in *alh-6* mutant day one adult males compared to age-matched WT males. (E) Spermatid size is reduced in *alh-6* mutant day one adult males as compared to age matched WT males fed HT115. (F) Sperm activation is impaired in *alh-6* mutant day one adult males relative to age-matched WT males fed HT115. Statistical comparisons of sperm number and size by unpaired t-test and sperm activation by Fisher's exact test. *, p<0.05; **, p<0.01; ***, p<0.001; ****, p<0.0001. All studies performed in at least biological triplicate; refer to ***Supplementary file 1*** for n for each comparison.

The online version of this article includes the following figure supplement(s) for figure 2:

**Figure supplement 1.** Activation defects of *alh-6* spermatids.

type (*Figure 2B*). To achieve motility, *C. elegans* spermatids must form a pseudopod which requires protease activation (*Ward et al., 1983*; *Figure 2—figure supplement 1B*). Sperm activation can be recapitulated in vitro by treatment of isolated spermatids with the *Streptomyces griseus* protease Pronase (*Shakes and Ward, 1989*). After 30 min of Pronase treatment, 80% of wildtype spermatids are fully activated, while a significantly reduced population of *alh-6* mutant spermatids mature over the same time period (*Figure 2C*). The reduction in activation, as measured by the presence of a fully extended pseudopod, in *alh-6* mutant spermatids is correlated with an increase in the number of cells observed at the normally transient intermediate stage of spermiogenesis characterized by the presence of 'spikes' (*Figure 2—figure supplement 1B–C*; *Shakes and Ward, 1989*). We observed a similar impairment in activation of *alh-6* mutant spermatids when treated with the cationic ionophore Monensin (*Figure 2—figure supplement 1D–E*), except that *alh-6* mutant spermatids were stalled at the 'protrusion' intermediate stage of spermiogenesis (*Nelson and Ward, 1980*). Future studies to reveal where and how mitochondrial proline catabolism integrates into specific stages of spermiogenesis will be of great interest (*Shakes and Ward, 1989*).

Interestingly, although sperm number was the same between WT and *alh-6* mutant males on the OP50 diet, sperm number was reduced in *alh-6* mutant males fed HT115 diet compared to age-matched WT males on the same diet (*Figure 2D*). We also noted that spermatids from *alh-6* mutant males raised on the HT115 diet were similarly defective in size and activation (*Figure 2E–F*). Taken together, although diet can influence sperm number, the reduction of sperm size and activation are likely contributors to the reduced fertility and competitive fitness in *alh-6* mutant males; which is independent of diet.

## Transcriptional signatures define temporal phenotypes of *alh-6* mutant animals

We first identified *alh-6* mutant in a screen for activators of the cytoprotective transcription factor SKN-1/NRF2 using *gst-4p::gfp* as a reporter (*Pang and Curran, 2014*; *Pang et al., 2014*). When activated, SKN-1 transcribes a variety of gene targets that collectively act to restore cellular homeostasis. However, this can come with an energetic cost with pleiotropic consequences (*Blackwell et al., 2015*; *Paek et al., 2012*; *Glover-Cutter et al., 2013*; *An and Blackwell, 2003*; *Lynn et al., 2015*; *Palikaras et al., 2015*; *Pang et al., 2014*; *Pang and Curran, 2014*). *alh-6* mutants have normal development, but display progeroid phenotypes towards the end of the normal reproductive span (*Pang and Curran, 2014*) indicating a temporal switch in phenotypic outcomes. We reasoned that the temporally controlled phenotypes in the *alh-6* mutants could be leveraged to identify potential mechanisms by which *alh-6* loss drives cellular dysfunction. As SKN-1 is activated in *alh-6* mutants after day 2 of adulthood (*Pang and Curran, 2014*), we defined genes that display differentially altered expression in the L4 developmental stage, when spermatogenesis occurs, as compared to day three adults (post SKN-1 activation). We performed RNA-Seq analyses of worms with loss of *alh-6* and identified 1935 genes in L4 stage animals and 456 genes in day three adult animals that are differentially expressed (+/- $Log_2$ (fold change), 0.05 FDR) (*Figure 3—figure supplement 1A–B*). Notably, the gene expression changes at these two life periods had distinct transcriptional signatures (*Figure 3A–B*). Because the loss of *alh-6* drives compensatory changes in normal cellular metabolism, which later in life results in the activation of SKN-1, we expected to identify significant changes in both metabolic genes and SKN-1 target genes. Supporting this hypothesis, the Gene Ontology (GO) terms most enriched include oxidoreductases and metabolic enzymes in L4 stage animals (*Figure 3A*) and SKN-1-dependent targets such as glutathione metabolism pathway genes in day three adults (*Figure 3B*). Importantly, our transcriptomic analysis recapitulated the temporally-dependent phenotypic outcomes resulting from *alh-6* loss; genes in the pseudopodium and germ plasm GO terms class displayed reduced expression in L4 *alh-6* mutant animals (*Figure 3A*), which include many genes in the major sperm protein (MSP) family that comprises 15% of total protein content in *C. elegans* sperm and impact sperm function (*Klass and Hirsh, 1981*). In contrast, genes in the muscle-specific GO term class displayed increased expression in day three adults (*Figure 3B*), which is when activation of the SKN-1 reporter is enhanced in the muscle of *alh-6* mutants (*Pang et al., 2014*). Taken together, the transcriptomic analysis of *alh-6* mutants is diagnostically relevant and informative for defining drivers of organism-level phenotypic changes in animals with altered proline catabolism.

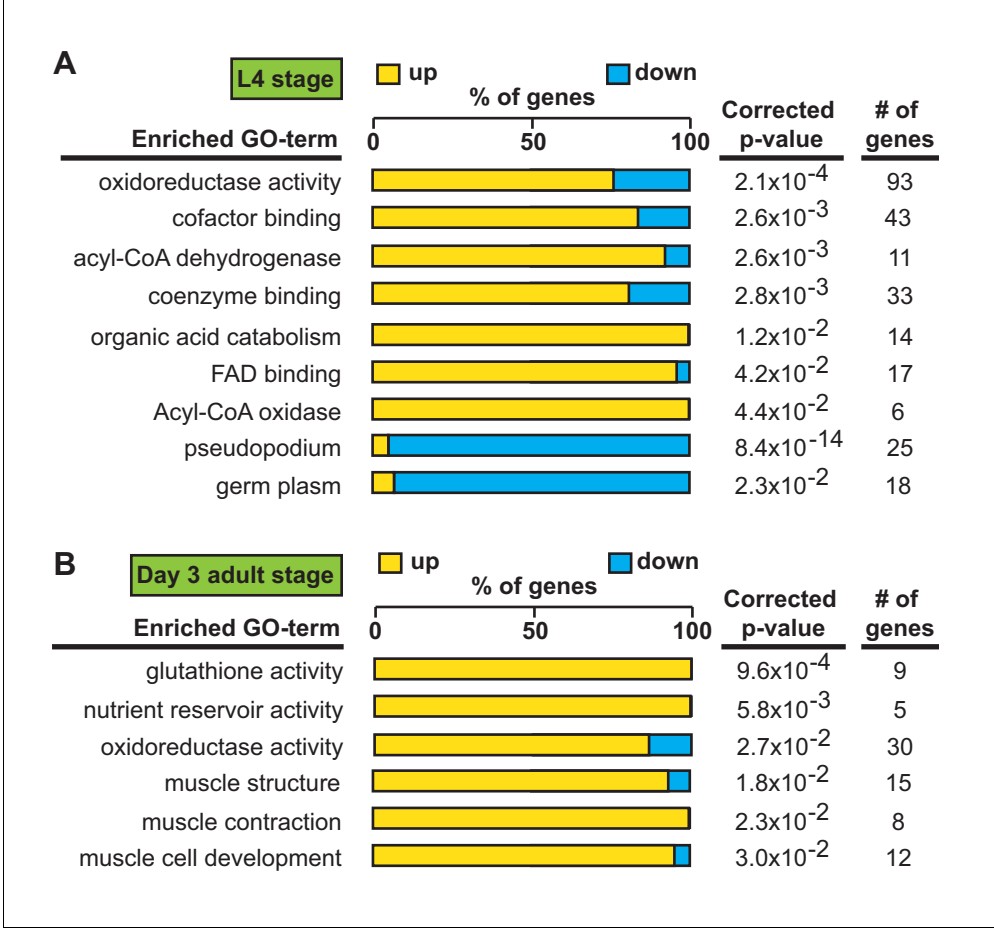

**Figure 3.** Transcriptional patterns define developmental- and adult-specific consequences to loss of *alh-6* activity. Gene Ontology (GO) term enrichment analysis of RNA-Seq data. (**A**) Transcriptional changes at L4 stage are enriched for metabolism and sperm-specific genes. (**B**) Transcriptional changes at day three adulthood are enriched for changes in glutathione activity, oxidoreductase activity, and muscle-specific genes. All studies performed in at least biological triplicate; refer to *Supplementary file 1* for n for each comparison.

The online version of this article includes the following figure supplement(s) for figure 3:

**Figure supplement 1.** RNA-Sequencing data of WT and *alh-6* hermaphrodites at L4 and day three adulthood.

## FAD mediates sperm functionality and competitive fitness

The strong enrichment of genes whose protein products utilize and/or bind cofactors or co-enzymes was intriguing as the maintenance of metabolic homeostasis and the redox state of the cell requires a sophisticated balance of multiple cofactors (*Figure 4A*). In fact, the proline catabolism pathway utilizes multiple cofactors to generate glutamate from proline; PRDH-1 uses FAD as a co-factor to convert proline to P5C while ALH-6 utilizes the reduction of NAD+ to convert P5C to glutamate. Additionally, in the absence of ALH-6, accumulation of P5C, the toxic metabolic intermediate of proline catabolism, drives the expression of pathways to detoxify P5C (oxidoreductases, P5C reductase, etc.) (*Figure 3*, *Figure 3—figure supplement 1C*). Although enzymes in the proline catabolism pathway utilize FAD as a cofactor, the transcriptional signature of the *alh-6* mutants includes the activation of multiple enzymes that utilize FAD, which drove the hypothesis that FAD levels might be altered in *alh-6* mutants. We measured FAD and found a significant reduction in *alh-6* mutant animals fed the OP50 diet at the L4 stage (*Figure 4B*) and a similar reduction in animals fed HT115 bacteria at L4 stage (*Figure 4C*). Differences in FAD levels were unremarkable in day three adult animals, when spermatogenesis has long since ended (*Figure 4—figure supplement 1A*). Based on this finding, we predicted that restoration of FAD levels might alleviate the sperm-specific phenotypes of *alh-6* mutants. Riboflavin is a precursor of FAD (*Figure 4D*) and dietary supplementation of

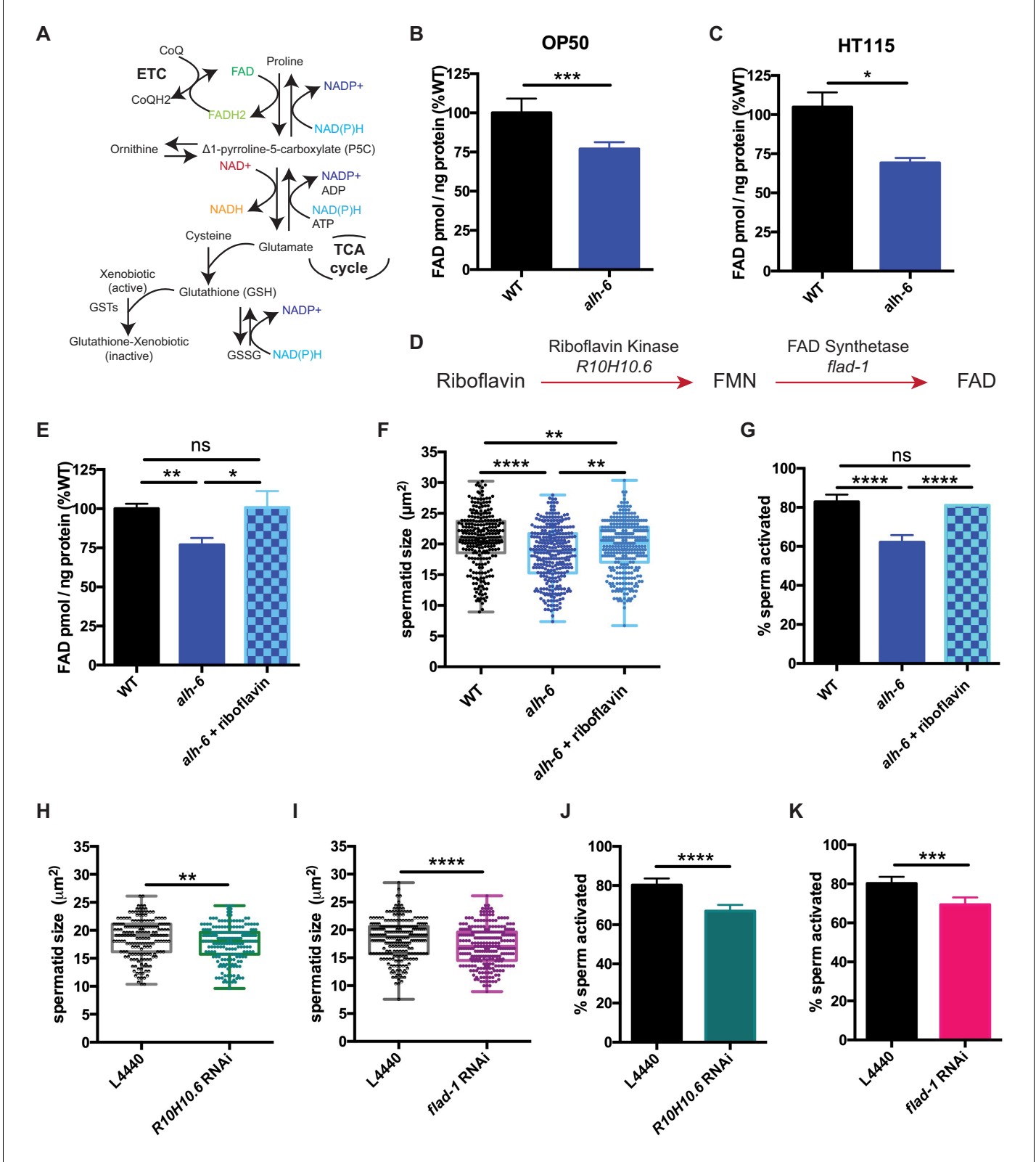

**Figure 4.** Loss of FAD homeostasis in *alh-6* mutants leads to sperm dysfunction. (A) Metabolic pathways utilize adenine dinucleotide cofactors to maintain redox balance in cells. (B–C) FAD+ levels are reduced in *alh-6* mutant animals fed OP50 (B) or HT115 (C) at the L4 developmental stage. (D) FAD biosynthetic pathway. (E–G) Dietary supplement of riboflavin restores FAD level (E), sperm size (F), and sperm activation (G) in *alh-6* mutants. (H–I) RNAi knockdown of *R10H10.6* (H) or *flad-1* (I) in WT males reduces their sperm size compared to L4440 vector control. (J–K) RNAi knockdown of
*Figure 4 continued on next page*

*Figure 4 continued*

*R10H10.6* (J) or *flad-1* (K) in WT males impairs sperm activation upon Pronase treatment. Statistical comparisons of sperm size by ANOVA. Statistical comparisons of activation by fisher's exact test with p-value cut-off adjusted by number of comparisons. *, p<0.05; **, p<0.01; ***, p<0.001; ****, p<0.0001. All studies performed in biological triplicate; refer to **Supplementary file 1** for n for each comparison.

The online version of this article includes the following figure supplement(s) for figure 4:

**Figure supplement 1.** Adenine nucleotide cofactor homeostasis is disrupted in *alh-6* mutants.

riboflavin has been shown to increase cellular FAD levels in wild-type animals (**Burch et al., 1956**; **Redondo et al., 1975**). Similarly, riboflavin supplementation to the OP50 diet of *alh-6* mutants restored FAD levels to wild-type levels (**Figure 4E**). We found that wild type hermaphrodites mated to *alh-6* mutant males fed a riboflavin supplemented diet produced significantly more total progeny than *alh-6* males fed the standard OP50 diet (**Figure 4—figure supplement 1B**). Moreover, riboflavin supplementation was sufficient to partially restore male sperm size (**Figure 4F**) and also rescued the impaired activation (**Figure 4G**) of male sperm in *alh-6* mutants. Riboflavin supplementation increases sperm size in WT males, but do not change sperm activation in WT males (**Figure 4—figure supplement 1C–D**).

We next asked whether FAD metabolism was required for proper sperm function. FAD can be synthesized de novo by a two-step enzymatic reaction where riboflavin is converted to FMN by Riboflavin Kinase/R10H10.6, which is subsequently converted to FAD by FAD Synthase/FLAD-1 (**Figure 4D**). We used RNA interference (RNAi) against *R10H10.6* or *flad-1* in wild-type male animals and measured sperm quality. Similar to *alh-6* mutant sperm, RNAi reduction of the FAD biosynthetic pathway decreased sperm size (**Figure 4H and I**, **Figure 4—figure supplement 1E–F**) and impaired sperm activation (**Figure 4J and K**, Figures **Figure 4—figure supplement 1E–F**).

NAD+ and NADH are also central adenine dinucleotide cofactors that play critical roles in metabolism and have received recent attention as a method to combat the decline seen in biological function with age (**Guarente, 2016**). As such, we also measured NAD and NADH levels, but found the ratio unremarkable between wild-type and *alh-6* mutant animals (**Figure 4—figure supplement 1G–I**). Taken together, these data suggest that loss of *alh-6* leads to a specific decrease in cellular FAD levels and that FAD is a critical cofactor that drives proper sperm function.

## Mitochondrial dynamics regulate spermatid function

Although there is a clear and documented role for mitophagy in the clearance of paternal mitochondria post-fertilization in *C. elegans*, the role(s) for mitochondrial dynamics and turnover in sperm function prior to zygote formation are unclear. We first examined mitochondrial dynamics in wild type spermatids by staining with the fluorescent mitochondrial-specific dye JC-1, and noted that each spermatid on average contained multiple discernable spherical mitochondria that are mostly not fused (**Figure 5A, B and E**). Previous studies in yeast and cultured mammalian cells have shown that when cells are exposed to mild stress, the initial response of mitochondria is to fuse in order to dilute damage (**Tondera et al., 2009**; **Gomes et al., 2011**; **Rambold et al., 2011**).

The mitochondrial specific dye JC-1 accumulates in mitochondria in a membrane potential-dependent manner, and as the concentration increases, its fluorescence switches from green to red emission. The accumulation of sufficient JC-1 molecules required for red emission is abolished by treatment with Carbonyl cyanide *m*-chlorophenyl hydrazone (CCCP), a chemical inhibitor of mitochondrial oxidative phosphorylation (**Figure 5—figure supplement 1A**). Therefore, a higher red-to-green fluorescence ratio in cells is indicative of healthier mitochondria species and as such, we characterized mitochondria with red JC-1 emission in our analyses of connectivity in spermatids. *alh-6* mutant spermatids have reduced red:green JC-1 fluorescence that indicates a lower mitochondrial membrane potential and an accumulation of unhealthy mitochondria (**Figure 5F**; **Smiley et al., 1991**). Moreover, *alh-6* mutant spermatids have mitochondria that were more interconnected (**Figure 5C–E**) as compared to wild type spermatids and a similar increase in connectivity was observed when mitochondria were visualized with the membrane potential-dependent mitochondrial dye Mitotracker Red CMXRos (**Figure 5—figure supplement 1B**). The increase in fused mitochondria in spermatids was also present in animals fed the HT115 diet, which further supports a diet-independent role for *alh-6* in the germline (**Figure 5—figure supplement 1C**).

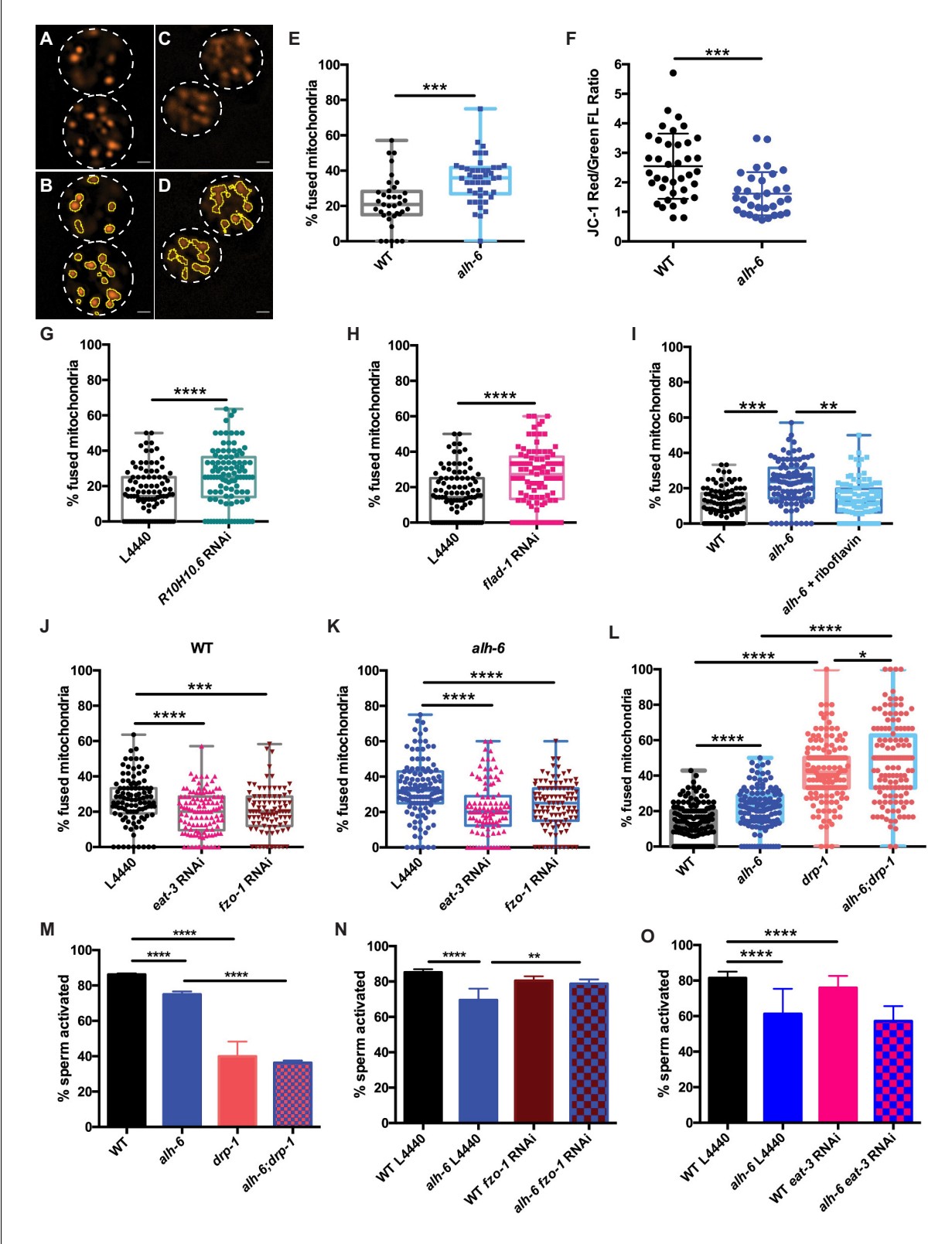

**Figure 5.** Mitochondrial dynamics drive sperm quality. (A–E) JC-1 dye stained mitochondria of WT (A–B), *alh-6* mutant (C–D); (B and D) are ImageJ detection of JC-1 stained sperm mitochondria area which are quantified in (E). (F) Mitochondria in *alh-6* mutant spermatids have reduced JC-1 red/green fluorescence ratio, indicating mitochondria depolarization. (G–H) RNAi knockdown of FAD biosynthetic pathway genes, *R10H10.6* (G) or *flad*-1 (H) increases mitochondrial fusion in WT spermatids. (I) Dietary supplement of FAD precursor riboflavin restores mitochondrial fusion in *alh-6* spermatids to

*Figure 5 continued on next page*

*Figure 5 continued*

WT level. (J–K) *eat-3* or *fzo-1* RNAi decreases mitochondrial fusion in both WT (J) and *alh-6* (K) mutant spermatids. (L) *drp-1* mutation increases mitochondrial fusion in both WT and *alh-6* spermatids. (M) *drp-1* mutation significantly impairs sperm activation in both WT and *alh-6* mutant spermatids. (N) *fzo-1* RNAi restores sperm activation in *alh-6* mutant. (O) *eat-3* RNAi reduces sperm activation in WT males but not *alh-6* males. Statistical comparisons of JC-1 Red/Green FL ratio by unpaired t-test. Statistical comparisons of mitochondria fusion by ANOVA. Statistical comparisons of sperm activation by Fisher's exact test with p-value cut-off adjusted by number of comparisons. *, $p<0.05$; **, $p<0.01$; ***, $p<0.001$; ****, $p<0.0001$. All studies performed in at least biological triplicate; refer to *Supplementary file 1* for n for each comparison.

The online version of this article includes the following figure supplement(s) for figure 5:

**Figure supplement 1.** *fzo-1* is involved in mitochondrial dynamics aberration in *alh-6* spermatids.

A connection between mitochondrial dynamics (fusion and fission) and FAD homeostasis has not been previously described. To understand this, we perturbed FAD biosynthesis pathway and then examined mitochondrial connectivity in spermatids. We first reduced FAD biosynthesis with RNAi targeting *R10H10.6* or *flad-1*, which resulted in more connected mitochondria that resembles the increased fusion in *alh-6* mutant spermatids that are under metabolic stress (*Figure 5G–H*, *Figure 4—figure supplement 1E–F*). In addition, increasing FAD levels by dietary supplementation of riboflavin, restored mitochondria in spermatids of *alh-6* animals to more wild-type-like distributions (*Figure 5I*), but did not change mitochondrial morphology in WT male spermatids (*Figure 5—figure supplement 1D*). Thus, the reduction of FAD in *alh-6* mutants, alters mitochondrial dynamics to a more fused and less punctate state. Therefore, the homeostatic control of FAD level is critical to maintain proper mitochondrial dynamics in sperm.

The role of mitochondrial dynamics in the maturation of sperm has not been studied; however recent work has revealed that the mitochondrial fusion and fission machinery are important for the elimination of paternal mitochondria post-fertilization (*Wang et al., 2016*). FZO-1 is required for proper fusion of the mitochondrial outer membrane while EAT-3/OPA1 regulates inner membrane fusion. In opposition to the activities of FZO-1 and EAT-3, DRP-1 is required for mitochondrial fission (*Smirnova et al., 2001*; *Lima et al., 2018*). The balance of this fusion and fission machinery in the upkeep of mitochondrial homeostasis allows cells to respond to changes in metabolic needs and external stress (*van der Bliek et al., 2017*; *Shaw and Nunnari, 2002*). RNAi of *fzo-1* or *eat-3* reduced mitochondrial fusion in wild-type male sperm (*Figure 5J* and *Figure 5—figure supplement 1E*) and suppressed the enhanced fusion observed in *alh-6* mutant spermatid mitochondria (*Figure 5K*); indicating mitochondrial fusion of both membranes is active in spermatids with impaired proline catabolism. We next examined spermatids from *drp-1* mutant animals and observed a greater level of mitochondrial fusion as compared to wild type and *alh-6* mutant spermatids (*Figure 5L*). We also observed a synergistic level of mitochondrial fusion in spermatids derived from *alh-6; drp-1* double mutants. This finding is consistent with previous studies in yeast which reveal that defects in fusion can be compensated for by changes in the rates of fission and vice versa (*Shaw and Nunnari, 2002*; *van der Bliek et al., 2017*). In support of our model where mitochondrial dynamics act as a major driver of the sperm-specific defects in *alh-6* mutants, we discovered that loss of *drp-1*, which results in increased mitochondrial fusion (like that observed in *alh-6* mutants), also reduces sperm activation (*Figure 5M*). Moreover, reducing *fzo-1* or *eat-3* does not alter activation in wild type sperm, while *fzo-1* but not *eat-3* RNAi restores activation in *alh-6* sperm (*Figure 5N–O* and *Figure 5—figure supplement 1E*), suggesting increased fusion mediated predominantly by *fzo-1* in *alh-6* sperm mitochondria is impairing proper function. We noted that *alh-6* mutant animals have an increased expression of *fzo-1* transcripts that is suggestive of a retrograde signaling response from the mitochondria (*Figure 5—figure supplement 1F*). Taken together, these data support a model where loss of mitochondrial proline catabolism induces mitochondrial stress, activating mitochondrial fusion, in order to dilute damage to preserve functional mitochondria at the cost of sperm function. These data also reveal a functional role for mitochondrial fusion and fission in spermatid development and sperm function.

## *alh-6* and FAD are cell autonomous regulators of sperm function

Signaling between germ and somatic cells can alter function in each cell type (*Ghazi et al., 2009*; *Curtis et al., 2006*; *Greenwald, 1989*; *Berman and Kenyon, 2006*; *Libina et al., 2003*; *Lin et al.,*

*2001*; *Hsin and Kenyon, 1999*). In light of the differences between somatic and germline phenotypes observed in *alh-6* mutant animals, we performed germline specific RNAi targeting *alh-6* to deduce whether the sperm defects observed were cell autonomous. Germline specific RNAi of *alh-6* in wild-type males was not sufficient to alter sperm size (*Figure 6A*), but did result in diminished sperm activation (*Figure 6B*,) and increased mitochondrial fusion in sperm (*Figure 6C*). Similarly, RNAi of *alh-6* only in the soma resulted in a minor reduction of spermatid size (*Figure 6—figure supplement 1A*), but did not phenocopy the impairment of sperm activation as observed in *alh-6* mutants (*Figure 6—figure supplement 1B*). Taken together, these findings suggest that somatic expression of *alh-6* can influence spermatid size while the influence of *alh-6* on spermatid activation is cell autonomous.

Next, we restored wild-type *alh-6* expression, only in the germline, in *alh-6* mutant animals, which restored sperm size in one of the two transgenic lines (*Figure 6D*), activation (*Figure 6E*) and mitochondrial dynamics (*Figure 6F*), as compared to non-transgenic siblings. We conclude that the effects of loss of *alh-6* on sperm function (activation and mitochondria) are cell autonomous because germline specific RNAi could phenocopy the sperm defects observed in whole animal loss of *alh-6*, while RNAi of *alh-6* only in the somatic tissues could not. In contrast, the effect of *alh-6* on sperm size is non-cell autonomous and requires somatic input (*Figure 6—figure supplement 1C–F*).

Since FAD functions in a variety of essential cellular processes, we next asked if proper sperm function required FAD homeostasis in germ cells. Similarly, we reduced *R10H10.6* or *flad-1* only in the germline, which phenocopies germline knockdown of *alh-6* on sperm size (*Figure 6G,J*), sperm activation (*Figure 6H,K*), and mitochondrial fusion in sperm (*Figure 6I,L*), as observed in whole animal RNAi of *flad-1* or *R10H10.6* (*Figure 4J–K* and *Figure 5G–H*).These results suggest that FAD functions similarly to *alh-6* in cell autonomously regulating sperm function (activation and mitochondrial dynamics), while affecting sperm size in a cell non-autonomous manner (*Figure 4H–I*). Taken together these data identify the importance of proline catabolism and FAD homeostasis in germ cells to maintain proper sperm function. In conclusion, our studies define mitochondrial proline catabolism as a critical metabolic pathway for male reproductive health.

## Discussion

Here we investigate the effects of disrupting mitochondrial proline catabolism through the loss of the mitochondrial enzyme gene *alh-6* and the resulting changes in FAD homeostasis, mitochondrial dynamics, and male fertility (*Figure 7*). We found that *alh-6* mutants show a reduction in brood size that is sexually dimorphic; defects in sperm function but not oocytes contribute to reduced hermaphrodite fertility. As societal factors continue to push individuals to wait longer to have children, the increase in paternal age is inversely correlated with proper sperm function and can give rise to fertility issues. Consequently, it is incumbent on future studies to elucidate how restoring and maintaining functional amino acid catabolism during aging in order to promote reproductive success.

Although *C. elegans* is a well-established organism for studying aging and reproduction, with several studies describing hermaphrodite reproductive senescence, many questions regarding the basis of male reproductive decline remain unanswered. Decades of work have shown that exposure to pollution, toxins, xenobiotics, and other ROS-inducing compounds can prematurely drive the loss of sperm function across species (*Agarwal et al., 2014*; *Wagner et al., 2018*; *Cocuzza et al., 2007*), but the impact that normal cellular metabolism plays on sperm function and the identification of specific molecules that can mediate sperm quality are not well-defined. In this study we characterized a new role for mitochondrial proline catabolism and FAD homeostasis in the maintenance of proper sperm function. Perturbation of this pathway, through mutation of *alh-6/ALDH4A1*, causes metabolic stress. Consequently, this perturbation leads to reduction of cellular FAD level and increases mitochondrial fusion in spermatids, which results in impaired sperm function and premature reproductive senescence.

Mutation in proline dehydrogenase (*PRODH*) in humans results in hyperprolinemia type I (HPI), while mutation in delta-1-pyrroline-5-carboxylate dehydrogenase (*ALDH4A1/P5CDH*) results in hyperprolinemia type II (HPII). This study reveals that in *C. elegans*, proline catabolism impacts several functional qualities of male sperm. Loss of proline catabolism results in smaller sperm with impaired activation, two qualities that directly impact competitive advantage. As such, proline biosynthesis, catabolism, and steady state concentrations must be tightly regulated, and the importance

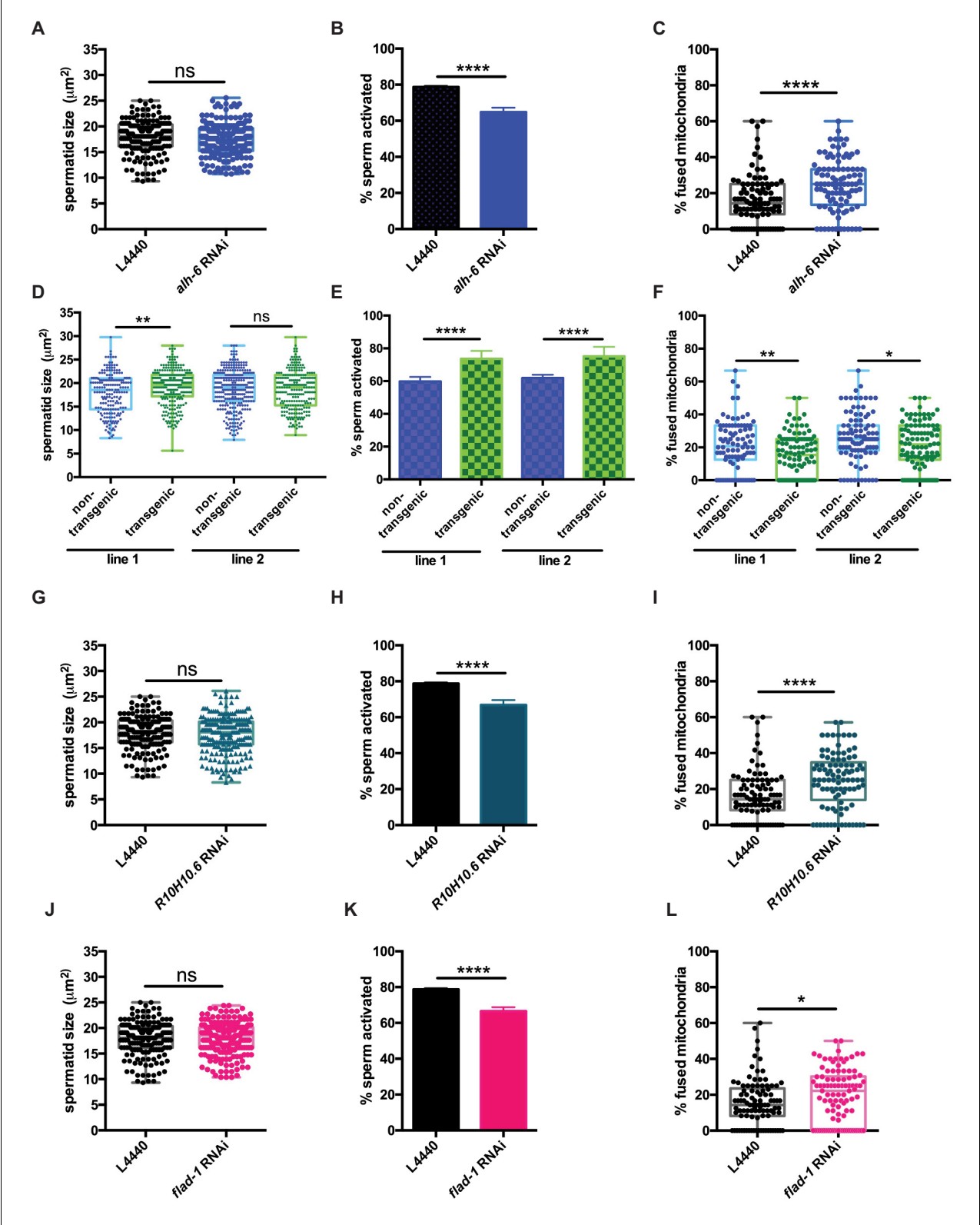

**Figure 6.** *alh-6* and FAD function cell autonomously in the germline to regulate sperm function. (A–C) Germline-specific RNAi of *alh-6* does not change sperm size (A), but does impair sperm activation (B) and increases mitochondrial fusion in sperm (C). (D–F) Germline-specific rescue of WT *alh-6* in *alh-6* mutant male animals increases sperm size (D) and restores activation (E) and mitochondrial dynamics (F). Statistical comparisons of sperm size and mitochondrial fusion in spermatids by unpaired t-test. Similarly, (G–L) germline-specific RNAi of *R10H10.6* and *flad-1* do not change sperm size (G,J),

*Figure 6 continued on next page*

*Figure 6 continued*

impair sperm activation (**H,K**), and increase mitochondrial fusion in sperm (**I,L**). Statistical comparisons of sperm activation by Fisher's exact test with p-value cut-off adjusted by number of comparisons. *, p<0.05; **, p<0.01; ***, p<0.001; ****, p<0.0001. All studies performed in biological triplicate; refer to *Supplementary file 1* for n for each comparison.

The online version of this article includes the following figure supplement(s) for figure 6:

**Figure supplement 1.** Germline expression of WT *alh-6* is sufficient to rescue sperm defect.

of proline in cellular homeostasis may help explain the transcriptional responses measured in animals with dysfunctional *alh-6*. Our data support a cell autonomous role for proline catabolism in sperm. However, although whole animal RNAi of *alh-6* closely phenocopies the *alh-6* mutant including reduced spermatid size, germline specific RNAi of *alh-6* did not significantly reduce the size of spermatids; perhaps suggesting a partial role for ALH-6 in somatic tissues for spermatid development, which is in line with recent studies in *C. elegans* describing soma to germline signaling in sperm activation (*Chavez et al., 2018*). Intriguingly, the impact of loss of *alh-6* is mostly independent of diet source, unlike the somatic phenotypes which are diet-dependent (*Pang and Curran, 2014*). The exception is sperm number in *alh-6* mutant animals on the HT115 diet, which appears to be diet-

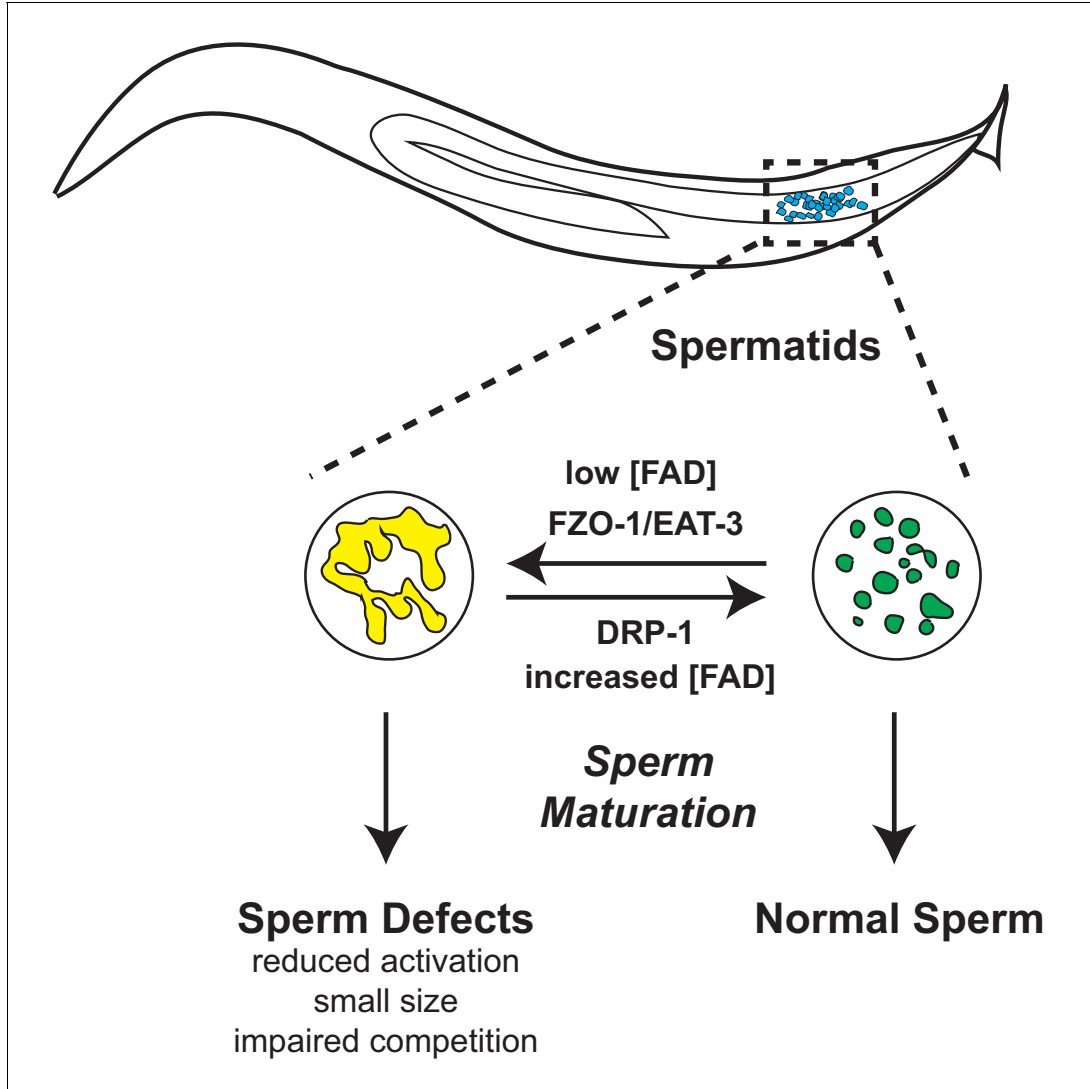

**Figure 7.** Model of *alh-6* and FAD mediated male reproductive senescence.

dependent (*Figure 2D*). WT males have more spermatids when fed the HT115 diet, as compared to WT animals fed OP50 diet, while *alh-6* mutants have the same number of spermatids on both diets.

Our previous work defined the age-dependent decline in function of somatic tissues, particularly muscle in animals lacking functional ALH-6 (*Pang and Curran, 2014*; *Pang et al., 2014*), which does not manifest until day 3 of adulthood. Our current study reveals that although somatic phenotypes in *alh-6* mutants are observed post-developmentally, the germline, or more specifically spermatids, are sensitive to loss of *alh-6* much earlier in development (phenotypes assayed at L4 or Day 1 of adulthood). Reproductive senescence is a field of growing significance as the number of couples that choose to delay having children increases. Importantly, although *alh-6* mutant sperm are impaired for competition, they remain viable for reproduction. This is similar to recent study on *comp-1,* a mutation which results in context-dependent competition deficit in *C. elegans* sperm (*Hansen et al., 2015*).

Recent studies have focused on the role of NAD+ metabolism in cellular health, while the impact of FAD has received less attention. FAD levels are diminished in *alh-6* animals specifically at the L4 stage when spermatogenesis is occurring. Riboflavin (Vitamin B$_2$) is a precursor to the FAD and FMN cofactors that are needed for metabolic reactions in order to maintain proper cellular function, like proline catabolism and mitochondrial oxidative phosphorylation. Despite its importance, humans, like *C. elegans*, lack a riboflavin biosynthetic pathway and therefore require riboflavin from exogenous sources (*Powers, 2003*). Insufficient intake can lead to impairment of flavin homeostasis, which is associated with cancer, cardiovascular diseases, anemia, neurological disorders, impaired fetal development, etc. (*Powers, 2003*). Our study suggests that riboflavin and FAD play critical roles in reproduction, specifically in germ cell development, as loss of FAD biosynthesis or loss of *alh-6* specifically in the germline recapitulates the sperm defects observed in whole animal knockdown or *alh-6* mutation. Importantly, these sperm-specific defects can be corrected by dietary supplementation of vitamin B$_2$, which in light of the exceptional conservation of mitochondrial homeostatic pathways, suggest the nutraceutical role vitamin B$_2$ could play in sperm health across species.

Our study also demonstrates that spermatids lacking *alh-6* have increased mitochondrial fusion; a perturbation at the mitochondrial organelle structure-level that contributes to the sperm-specific phenotypes observed. In addition to prior work showing *fzo-1/MFN1/MFN2* and *drp-1/DRP-1* to be important for mitochondrial elimination post-fertilization (*Wang et al., 2016*), our work reveals that mitochondrial fission and fusion machinery are present and active in spermatids and that perturbation of these dynamics can affect sperm maturation and competitive fitness. Future work to define how *alh-6* spermatids use mitophagy, which can clear damaged mitochondria, will be of interest. In conclusion, our work identifies proline metabolism as a major metabolic pathway that can impact sperm maturation and male reproductive success. Moreover, these studies identify specific interventions to reverse the redox imbalance, cofactor depletion, and altered mitochondria dynamics, all of which play a part in sperm dysfunction resulting from proline metabolism defects.

# Materials and methods

## Key resources table

| Reagent type (species) or resource | Designation | Source or reference | Identifiers | Additional information |
|---|---|---|---|---|
| Strain (*C. elegans*) | N2 | *Caenorhabditis* Genetics Center (CGG) | | Laboratory reference strain (wild type) |
| Strain (*C. elegans*) | SPC321 | PMID: 24440036 | | Genotype: *alh-6(lax105)* |
| Strain (*C. elegans*) | SPC326 | PMID: 24440036 | | *alh-6p::alh-6::gfp* |
| Strain (*C. elegans*) | SPC447 | This paper | | Genotype: *alh-6(lax105); laxEx025(pie-1p::alh-6; myo-2p::rfp;myo-3p::rfp;rab-3p::rfp)* |

*Continued on next page*

*Continued*

| Reagent type (species) or resource | Designation | Source or reference | Identifiers | Additional information |
|---|---|---|---|---|
| Strain (*C. elegans*) | SPC455 | This paper | | Genotype: *alh-6(lax105); laxEx033(pie-1p::alh-6; myo-2p::rfp;myo-3p:: rfp;rab-3p::rfp)* |
| Strain (*C. elegans*) | SPC473 | This paper | | Genotype: *alh-6(lax105); laxEx051(pie-1p::alh-6; myo-2p::rfp;myo-3p:: rfp;rab-3p::rfp)* |
| Strain (*C. elegans*) | CL2166 | *Caenorhabditis* Genetics Center (CGG) | | Genotype: *gst4-p::gfp* |
| Strain (*C. elegans*) | SPC223 | PMID: 24440036 | | Genotype: *alh-6(lax105); gst-4p::gfp* |
| Strain (*C. elegans*) | DCL569 | *Caenorhabditis* Genetics Center (CGG) | | Genotype: *[mkcSi13(sun-1p::rde-1::sun-1 3'UTR + unc-119(+)) II; rde-1(mkc36) V* |
| Strain (*C. elegans*) | CU6372 | *Caenorhabditis* Genetics Center (CGG) | | Genotype: *drp-1(tm1108)* |
| Strain (*C. elegans*) | GR1948 | PMID: 24684932 | | Genotype: *mut-14 (mg464);smut-1(tm1301) V.* |
| Chemical compound, drug | Riboflavin | Millipore Sigma | R9504 | Concentration used: 2.5 mM |
| Commercial Assay or kit | FAD Colorimetric/ Fluorometric Assay Kit | BioVision | K357 | |
| Commercial Assay or kit | NAD/NADH Quantification Colorimetric Kit | BioVision | K337 | |
| Chemical compound, drug | Pronase | Millipore Sigma | P8811 | Concentration used: 200 ug/mL |
| Chemical compound, drug | eBioscience Monensin Solution (1000X) | Thermo Fisher Scientific | 00-4505-51 | Concentration used: 100 nM |
| Chemical compound, drug | MitoProbe JC-1 Assay Kit | Thermo Fisher Scientific | M34152 | Concentration used: JC-1 15 uM, CCCP 50 uM |
| Chemical compound, drug | MitoTracker Red CMXRos | Thermo Fisher Scientific | M7512 | Concentration used: 100 uM dried on plate |
| Software | GraphPad Prism | GraphPad Prism (https://graphpad.com) | RRID:SCR_015807 | Version 6 |
| Software | ImageJ | ImageJ (http://imagej. nih.gov/ij/) | RRID:SCR_003070 | |

## *C. elegans* strains and maintenance

*C. elegans* were cultured using standard techniques at 20°C. The following strains were used: wild type (WT) N2 Bristol, SPC321[*alh-6(lax105)*], SPC326[*alh-6p::alh-6::gfp*], SPC447[*alh-6(lax105); laxEx025(pie-1p::alh-6;myo-2p::rfp;myo-3p::rfp;rab-3p::rfp)*], SPC455[*alh-6(lax105);laxEx033(pie-1p:: alh-6;myo-2p::rfp;myo-3p::rfp;rab-3p::rfp)*], SPC473[*alh-6(lax105);laxEx051(pie-1p::alh-6;myo-2p::rfp; myo-3p::rfp;rab-3p::rfp)*], CL2166[*gst4-p::gfp*], SPC223[*alh-6(lax105);gst-4p::gfp*], DCL569[*mkcSi13 (sun-1p::rde-1::sun-1 3'UTR + unc-119(+)) II; rde-1(mkc36) V*], CU6372[*drp-1(tm1108)*], and GR1948 [*mut-14(mg464);smut-1(tm1301) V*]. Double and triple mutants were generated by standard genetic techniques. *E. coli* strains used were as follows: B Strain OP50 (*Brenner, 1974*) and HT115(DE3) [F⁻mcrA mcrB IN(rrnD-rrnE)one lambda⁻ rnc14::Tn10 λ(DE3)](*Timmons et al., 2001*). For dietary supplement assays, riboflavin was added to the NGM plate mix to final concentration 2.5 mM.

## RNAi-based experiments

RNAi experiments were done using HT115-based RNAi (*Timmons et al., 2001*), which yielded similar results as OP50 RNAi *E. coli* B strain as described in *Dalton and Curran (2018)*. All strains were

adapted to diets for at least three generations and strains were never allowed to starve. All RNAi clones were sequenced prior to use and RNAi knockdown efficiency measured. RNAi cultures were seeded on IPTG plates and allowed to induce overnight prior to dropping eggs on them for experiments.

## Microscopy

Zeiss Axio Imager and ZEN software were used to acquire all images used in this study. For GFP reporter strains, worms were mounted in M9 with 10 mM levamisole and imaged with DIC and GFP filters. For sperm number, assay samples were imaged with DIC and DAPI filters in z-stacks. For sperm size and activation assays, dissected sperm samples were imaged at 100x with DIC filter on two different focal planes for each field to ensure accuracy. For sperm mitochondria assays, dissected sperm samples were imaged at 100x with DIC, GFP, and RFP filters in z-stacks to assess overall mitochondria content within each spermatid.

## Fertility assay

Worms were treated with alkaline hypochlorite and eggs were allowed to hatch overnight. The next day, synchronized L1 larvae were dropped on NGM plates seeded with either OP50 or HT115. 48 hr later, at least ten L4 hermaphrodites for each genotype were singled onto individual plates and moved every 12 hr until egg laying ceased. Progeny were counted 48 hr after the singled hermaphrodite was moved to a different plate. Plates were counted twice for accuracy.

## Mated reproductive assay

Males were synchronized by egg laying, picked as L4 larvae for use as young adults for mating experiments. Singled L4 stage hermaphrodites were each put on a plate with 30 ul of OP50 seeded in the center together with three virgin adult males. 24 hr post-mating, males were removed, and each hermaphrodite was moved to a new plate every 24 hr until egg laying ceased. Progeny were counted 48 hr after the hermaphrodite was moved from the plate. For sperm competition assay, progeny with GFP fluorescence were counted from the cohort. Plates were counted twice for accuracy.

## Cofactor measurements

Worms were treated with alkaline hypochlorite and eggs were allowed to hatch overnight. The next day, synchronized L1s were dropped on NGM plates with or without supplement seeded with 25X concentrated OP50. FAD levels are measured following directions in FAD Colorimetric/Fluorometric Assay Kit (K357) from BioVision. NAD/NADH levels are measured following directions in NAD/NADH Quantification Colorimetric Kit (K337).

## Sperm number assay

Worms were treated with alkaline hypochlorite and eggs were allowed to hatch overnight. The next day, synchronized L1s were dropped on NGM plates with the indicated food source. At 48 hr (L4 developmental stage) males were isolated to new plates. 72 hr post-drop, day one adult virgin male animals were washed 3x with 1xPBST, fixed with 40% 2-propanol, and stained with DAPI for 2 hr. Samples were washed for 30 min with PBST, mounted with Vectashield mounting medium, and covered with coverslip to image. Spermatids in the seminal vesicle were counted through all planes in z-stack.

## Sperm size assay

Males were isolated at L4 stage 24 hr before assay. For each strain, five day one adult males were dissected in 35 μL pH 7.8 SM buffer (50 mM HEPES, 50 mM NaCl, 25 mM KCl, 5 mM CaCl$_2$, 1 mM MgSO$_4$, 10 mM dextrose) to release spermatids, which were immediately imaged.

## Sperm activation with pronase and monensin

Males were isolated at L4 stage 24 hr before assay. For each strain, five day one adult males were dissected in 35 μL pH 7.8 SM buffer (50 mM HEPES, 50 mM NaCl, 25 mM KCl, 5 mM CaCl$_2$, 1 mM MgSO$_4$, 1 mg/ml BSA) supplemented with either 200 μg/mL Pronase (Millipore Sigma) or 100 nM

Monensin (Thermo Fisher Scientific 00-4505-51) to release spermatids. Another 25 ul of the same solution was added and the spermatids were incubated at RT for 30 min for activation to occur before imaging.

### Sperm mitochondria staining

Males were isolated at L4 stage 24 hr before assay. For each strain, five day one adult males were dissected in 35 µL pH 7.8 SM buffer (50 mM HEPES, 50 mM NaCl, 25 mM KCl, 5 mM $CaCl_2$, 1 mM $MgSO_4$, 1 mg/ml BSA) with JC-1(Thermo Fisher Scientific M34152) added to 15 µM final concentration. Another 25 ul of the same solution was added and the spermatids were incubated at RT for 10 min. The slide was washed three times with 100 ul SM buffer before imaging. For carbonyl cyanide *m*-chlorophenyl hydrazine (CCCP) uncoupler control in JC-1 staining experiment, 50 uM final concentration was used in staining solution. For staining with MitoTracker Red CMXRos (Thermo Fisher Scientific M7512), stock solution was diluted to 100 uM final concentration in M9 and 50 ul of this solution was applied on top of a spot of 50 uL 25X concentrated OP50 seeded on a NGM plate. Solution was allowed to dry on the plate before L4 virgin males were moved onto the food spot. Animals were allowed to stain overnight (18–24 hr) and dissected next day in SM buffer for spermatids to image.

### RNA-Sequencing

Worms were egg prepped and eggs were allowed to hatch overnight. The next day, synchronized L1s were dropped on NGM plates seeded with 25X concentrated OP50. 48 and 120 hr post drop, L4 animals and day three adult animals, respectively, were washed three times with M9 and frozen in TRI Reagent at −80°C. Animals were homogenized and RNA extraction was performed following the protocol in Zymo Direct-zol RNA Isolation Kit. RNA samples were sequenced and analyzed by Novogene.

### Statistical analysis

Data are presented as mean ± SEM. Comparisons and significance were analyzed in Graphpad Prism 7. Comparisons between two groups were done using Student's Test. Comparisons between more than two groups were done using ANOVA. For sperm activation assays, Fisher's Exact Test was used and p-values are adjusted for multiple comparisons. *p<0.05 **p<0.01 ***p<0.001 ****<0.0001.

## Acknowledgements

We thank N Mih, K Han, and L Thomas for technical assistance; H Dalton, A Hammerquist, N Stuhr, W Escorcia, and J Nhan for critical reading of the manuscript; C Phillips for the soma-restricted RNAi strain GR1948; and D Chavez for protocol on MitoTracker Red staining. Some strains were provided by the CGC, which is funded by the NIH Office of Research Infrastructure Programs (P40 OD010440). This work was funded by the NIH (R01GM109028, R01AG058610, to SPC, T32AG000037 to DLR, and T32GM118289 to CDT), and the American Federation of Aging Research (C-AY and SPC).

## Additional information

### Funding

| Funder | Grant reference number | Author |
| --- | --- | --- |
| National Institutes of Health | GM109028 | Sean P Curran |
| National Institutes of Health | AG058610 | Sean P Curran |
| National Institutes of Health | AG063947 | Sean P Curran |
| National Institutes of Health | AG000037 | Dana L Ruter |
| National Institutes of Health | GM118289 | Christian D Turner |
| American Federation for Aging Research | | Chia-An Yen Sean P Curran |

The funders had no role in study design, data collection and interpretation, or the decision to submit the work for publication.

## Author contributions
Chia-An Yen, Data curation, Formal analysis, Validation, Investigation, Visualization, Methodology, Writing - review and editing; Dana L Ruter, Investigation; Christian D Turner, Resources, Validation, Investigation, Methodology; Shanshan Pang, Investigation, Methodology; Sean P Curran, Conceptualization, Resources, Data curation, Formal analysis, Supervision, Funding acquisition, Validation, Investigation, Methodology, Writing - original draft, Project administration, Writing - review and editing

## Author ORCIDs
Sean P Curran (iD) https://orcid.org/0000-0001-7791-6453

## Decision letter and Author response
Decision letter https://doi.org/10.7554/eLife.52899.sa1
Author response https://doi.org/10.7554/eLife.52899.sa2

# Additional files

## Supplementary files
• Supplementary file 1. Sample size and replicate number for all experiments.
• Transparent reporting form

## Data availability
All relevant data has been provided. RNA-Seq data are deposited in GEO database (GSE121920).

The following dataset was generated:

| Author(s) | Year | Dataset title | Dataset URL | Database and Identifier |
|---|---|---|---|---|
| Yen C-A, Curran SP | 2020 | Loss of mitochondrial proline catabolism depletes FAD, impairing sperm function, and male reproductive advantage | https://www.ncbi.nlm.nih.gov/geo/query/acc.cgi?acc=GSE121920 | NCBI Gene Expression Omnibus, GSE121920 |

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
