## [Decision Letter]

**Acceptance summary:**

This study defines a cell autonomous role of mitochondria proline catabolism and FAD homeostasis in sperm functionality and competitive fitness. These interesting findings help us to understand how male sperm quality is controlled.

**Decision letter after peer review:**

[Editors’ note: the authors submitted for reconsideration following the decision after peer review. What follows is the decision letter after the first round of review.]

Thank you for submitting your work entitled "Loss of mitochondrial proline catabolism depletes FAD, impairing sperm function, and male reproductive advantage" for consideration by *eLife*. Your article has been reviewed by two peer reviewers, one of whom is a member of our Board of Reviewing Editors, and the evaluation has been overseen by a Senior Editor. The reviewers have opted to remain anonymous.

Our decision has been reached after consultation between the reviewers. Based on these discussions and the individual reviews below, we regret to inform you that your work will not be considered further for publication in *eLife*.

The reviewers agreed that the findings of this study, namely that a perturbation of mitochondrial proline catabolism impairs sperm quality and competitive fitness, are novel and interesting. However, this study falls short of providing mechanistic information to meet the standards for *eLife*. A substantial amount of work needs be conducted to address the concerns raised by the reviewers.

*Reviewer #1:*

In this manuscript, Yen et al. found that perturbation of mitochondrial proline catabolism impairs sperm quality and competitive fitness. The authors showed that loss of function of *alh-6*, encoding a mitochondrial enzyme involved in proline metabolism, leads to oxidative stress, FAD depletion and excess mitochondrial fusion. These pleiotropic defects reduce sperm quality. This study defines a role of mitochondria proline catabolism and FAD homeostasis in sperm functionality and competitive fitness. Previous studies have proved the importance of mitochondrial activity and ROS in male sperm function. Altered mitochondrial structure and increased level of ROS in *alh-6* loss of function mutants has also been shown by the same group. In this study, the authors observed a correlation of FAD levels and ROS with spermatid size and activation. However, no mechanistic insights have been shown. Whether levels of FAD and ROS act autonomously in sperm or in other tissues to regulate sperm functionality has not been addressed. In general, this study fails to meet the stringent requirements for publication in *eLife*.

1) The authors showed that levels of FAD are reduced in *alh-6* mutants. Levels of FAD in sperm could be determined.

2) Addition of riboflavin or antioxidant NAC restores spermatid size and activation in *alh-6* mutants. Do they affect sperm function in wild type animals or in other sperm defective mutants? Do they act autonomously to regulate sperm function?

*Reviewer #2:*

The paper by Yen et al. describes that loss of *alh-6* leads to P5C accumulation and FAD depletion, which impair sperm functions in *C. elegans*. The authors further found that blocking P5C generation, or supplement with FAD or NAC, or modulating mitochondrial dynamics restored sperm functions in *alh-6* mutants. These findings suggest a requirement for proper P5C catabolism in sperm quality control. Nevertheless, the causal effects of FAD depletion or P5C accumulation on ROS change and mitochondrial defects in the absence of *alh-6* need to be clarified for the paper to be considered for publication in *eLife*.

1) The title needs to be reconsidered, because blocking the first step of proline catabolism by loss of *prdh-1* in fact suppressed the sperm defects in *alh-6* mutants.

2) The RNA-seq data revealed genes that are likely involved in FAD binding (Figure 3), what are the effects on FAD, mitochondria, and sperm activities when these genes (17) are inactivated (in the backgrounds of WT and *alh-6*)?

3) Is the reduction of FAD in *alh-6* mutants diet-specific? Does *alh-6* mutation influence NAD+ level? Does supplement with FAD rescue the mitochondrial defects (both somatic and sperm) in *alh-6* mutants? This could distinguish whether the decrease in FAD or accumulation of P5C drives mitochondrial and sperm defects.

4) Is the FAD level restored to WT in *alh-6;prdh-1* double mutants? This is important to draw the conclusion that PRDH-1 continues to deplete FAD in the absence of *alh-6* (subsection “FAD mediates sperm functionality and competitive fitness”).

5) The subtitle "loss of cellular proline catabolism is not causal for sperm defects in *alh-6* mutants" needs to be reconsidered. It appears contradictory to the title of the manuscript.

6) The authors proposed that accumulation of P5C could account for ROS change that leads to sperm defects in *alh-6* mutants. Experimental data should be provided on restoration of ROS levels by supplement with NAC and FAD.

7) Are mitochondrial defects seen in *alh-6* sperms diet-specific? The authors showed that partial inactivation of *fzo-1* restored sperm activation, how about RNAi of *eat-3*? Does RNAi of *fzo-1* and *eat-3* change the ROS levels in *alh-6* mutant sperms?

8) The current model (Figure 6N) failed to define the relationships of P5C accumulation, FAD depletion, ROS change, mitochondrial dynamics and sperm quality. A better model needs to be suggested.

[Editors’ note: further revisions were suggested prior to acceptance, as described below.]

Thank you for submitting your article "Loss of flavin adenine dinucleotide (FAD) impairs sperm function and male reproductive advantage in *C. elegans*" for consideration by *eLife*. Your article has been reviewed by three peer reviewers, one of whom is a member of our Board of Reviewing Editors, and the evaluation has been overseen by Didier Stainier as the Senior Editor. The reviewers have opted to remain anonymous.

The reviewers have discussed the reviews with one another and the Reviewing Editor has drafted this decision to help you prepare a revised submission.

The revision has much improved the manuscript. Please address the concerns raised by reviewers #2 and # 3, which are pretty straightforward.

Reviewer #1:

My concerns raised in the initial round of reviewing process have been satisfactorily addressed. The authors provide new data to show that FAD acts autonomously in sperm to regulate sperm functionality. It is acceptable now.

Reviewer #2:

The manuscript by Yen et al. reports that impairment of mitochondrial proline metabolism caused by loss of *alh-6* leads to reduction of FAD levels, alteration of mitochondrial dynamics in sperm, and age-related pleiotropic consequences on sperm size and activity. They further found that inactivation of *alh-6* or FAD synthesis genes in the germline is sufficient to recapitulate the sperm defects as in whole animals with *alh-6* loss of function. They suggest that mitochondrial proline metabolism and FAD homeostasis play a cell autonomous role to maintain sperm function. These findings are very interesting, especially, by revealing the requirement of FAD for mitochondrial dynamics and sperm function. The manuscript is greatly improved compared with the previous one.

1) It is interesting that loss of ALH-6, but not PRDH-1 which uses FAD as cofactor, caused reduction of FAD levels. What are the possible biochemical explanations? Can this be experimentally tested?

2) The authors performed germline-specific RNAi to inactivate *alh-6* and FAD synthesis genes, which induced sperm defects as in *alh-6* mutant animals. In addition, expression of ALH-6 in the germline rescued the sperm defects. Thus, the conclusion is drawn that the effects of *alh-6* loss on sperm function are cell autonomous. This seems not to be sufficient. Will overexpression of ALH-6 in somatic tissues, e.g., the intestine, rescue the sperm defect in *alh-6* mutants?

Reviewer #3:

In this manuscript, Yen et al. found that loss of mitochondrial proline catabolism impairs sperm quality (size and activation) and male sperm competitiveness over hermaphrodite sperm. They showed that reduction of FAD levels lead to more mitochondria fusion, smaller sperm size, and less sperm activation by artificial sperm activator Pronase. In mature spermatids, both transcription and translation are quiescent, therefore, mitochondrial proline catabolism involved in the sperm quality control is very intriguing. It has been long known that male sperm size is larger, correlated with their superior competitiveness over hermaphrodite sperm during a regular crossing. This paper however does not provide a mechanistic insights about the link between the physical size of sperm and its mitochondrial proline catabolism, how FAD affects cell size.

1) Though Pronase (a mixture of proteases which cleave any surface protein on the cell) could be used to assess wild type sperm activation, Monensin is a better artificial sperm activation as it can make activated spermatozoa to crawl. At least, the authors need to check at which step mutant sperm failed to activate. The crossing data (in vivo sperm activation in Figure 1F) are inconsistent to the rest mutant sperm activation data.

2) JC-1 staining (Figure 5A-D) was employed extensively in this study to show the mitochondrial fusion or not. This reviewer suggests confirming that with a GFP-tagged mitochondrial protein such as TOM20 to avoid any bias for imaging collection and analysis.

3) Figure 1—figure supplement 1D should use a better image instead of the current one.

---

## [Author Response]

[Editors’ note: the authors resubmitted a revised version of the paper for consideration. What follows is the authors’ response to the first round of review.]

Reviewer #1:In this manuscript, Yen et al. found that perturbation of mitochondrial proline catabolism impairs sperm quality and competitive fitness. The authors showed that loss of function of alh-6, encoding a mitochondrial enzyme involved in proline metabolism, leads to oxidative stress, FAD depletion and excess mitochondrial fusion. These pleiotropic defects reduce sperm quality. This study defines a role of mitochondria proline catabolism and FAD homeostasis in sperm functionality and competitive fitness. Previous studies have proved the importance of mitochondrial activity and ROS in male sperm function. Altered mitochondrial structure and increased level of ROS in alh-6 loss of function mutants has also been shown by the same group. In this study, the authors observed a correlation of FAD levels and ROS with spermatid size and activation. However, no mechanistic insights have been shown. Whether levels of FAD and ROS act autonomously in sperm or in other tissues to regulate sperm functionality has not been addressed. In general, this study fails to meet the stringent requirements for publication in eLife.

We appreciate this feedback. By using germline specific RNAi of FAD biosynthesis genes in combination with germline specific recue of *alh-6*, we have determined the cell autonomous nature of FAD loss and *alh-6* activity in germ cells (Figure 6A-L). The generation and validation of these transgenic and the subsequent analyses of all sperm quality assays are the reasons this resubmission took 10 months to complete. We agree that this was an essential piece of the story that was missing and greatly enhances our study.

1) The authors showed that levels of FAD are reduced in alh-6 mutants. Levels of FAD in sperm could be determined.

We would love to measure FAD in sperm. However, the assays we used (we have tried several commercial kits) all require a minimum of 10^6^ and in some cases 10^8^ cells to meet sensitivity thresholds. Attaining this number of *C. elegans* spermatids while maintaining integrity of FAD is not possible; although we have tried to get close to this number and assay for FAD which was below detectable range. Similarly, measurements in dissected male gonads was also below detection limit.

As an alternative approach, we now show that reducing the expression of the FAD biosynthesis enzymes in germ cells phenocopies the sperm defects observed in the *alh-6* mutants.

Moreover, reducing *alh-6* expression only in germ cells, also recapitulates the sperm defects in *alh-6* mutants (Figure 6A-I).

We also show that the loss of *alh-6* does not affect NAD/NADH (Figure 4—figure supplement 1G-I), thus the effects on FAD are specific. Due to this specificity and novelty of FAD in affecting sperm function, we decided to focus on describing this mechanism in this manuscript.

2) Addition of riboflavin or antioxidant NAC restores spermatid size and activation in alh-6 mutants. Do they affect sperm function in wild type animals or in other sperm defective mutants? Do they act autonomously to regulate sperm function?

We have tested the impact of NAC on WT sperm and we found that there were no changes, suggesting the effect of ROS on *alh-6* sperm defects is specific.

WT males fed riboflavin supplement have increased sperm size, but activation and mitochondrial distributions remain unchanged (new Figure 4—figure supplement 1C-D, Figure 5—figure supplement 1D). The rescue of all sperm defects in *alh-6* mutants with dietary riboflavin supplement suggests FAD reduction is causal to these defects (Figure 4E-G, Figure 5I).

The idea that riboflavin could act as a general therapeutic for defective sperm was intriguing. There are several sperm mutants in *C. elegans* with known defects in spermatogenesis. However, none of these mutants have known functions in mitochondrial metabolism. Nevertheless, we have tested *spe-10* mutant with NAC and riboflavin dietary supplements, both of which do not rescue its severe sperm activation defect. Although a negative result, we would be happy to include this data (see Author response image 1, n = 40-80 individual spermatids) as it suggests that riboflavin treatment is specific to the sperm defects in *alh-6* mutants, supporting the causal role of reduced FAD.

Reviewer #2:The paper by Yen et al. describes that loss of alh-6 leads to P5C accumulation and FAD depletion, which impair sperm functions in *C. elegans*. The authors further found that blocking P5C generation, or supplement with FAD or NAC, or modulating mitochondrial dynamics restored sperm functions in alh-6 mutants. These findings suggest a requirement for proper P5C catabolism in sperm quality control. Nevertheless, the causal effects of FAD depletion or P5C accumulation on ROS change and mitochondrial defects in the absence of alh-6 need to be clarified for the paper to be considered for publication in eLife.

We agree! By using germline specific RNAi of FAD biosynthesis genes in combination with germline specific recue of *alh-6*, we have determined the cell autonomous nature of FAD loss and *alh-6* dependent roles in germ cells (new Figure 6A-L). We agree that this was an essential piece of the story that was missing.

1) The title needs to be reconsidered, because blocking the first step of proline catabolism by loss of prdh-1 in fact suppressed the sperm defects in alh-6 mutants.

We agree that our title needed to be more accurate. Due to the specificity and novelty of FAD in affecting sperm function, we have decided to focus on describing this mechanism in this paper. As such, we have changed our title to emphasize the importance of FAD.

2) The RNA-seq data revealed genes that are likely involved in FAD binding (Figure 3), what are the effects on FAD, mitochondria, and sperm activities when these genes (17) are inactivated (in the backgrounds of WT and alh-6)?

This is an interesting question. Inactivation of any one of these is unlikely to have an effect, but the combined actions of all is what likely results in the reduction of FAD level in *alh-6* animal at larval stage 4 when spermatogenesis occurs (Figure 4B-C).

We now show that germ cell-specific RNAi of FAD biosynthesis or germ cell-specific RNAi of *alh-6* drives similar activation impairment and altered mitochondrial dynamics as observed in the *alh-6* mutants (new Figure 6A-I). Moreover, restoring WT *alh-6* expression in the germline is sufficient to rescue sperm defects in *alh-6* mutants (new Figure 6J-L).

3) Is the reduction of FAD in alh-6 mutants diet-specific? Does alh-6 mutation influence NAD+ level? Does supplement with FAD rescue the mitochondrial defects (both somatic and sperm) in alh-6 mutants? This could distinguish whether the decrease in FAD or accumulation of P5C drives mitochondrial and sperm defects.

These are great questions that we now clarify in our study. In the revised manuscript we include all data on HT115 and OP50 for all sperm phenotypes This data can be found in the following new figure panels:

– Sperm number (Figure 2A and 2D)

– Sperm size (Figure 2B and 2E)

– Sperm activation (Figure 2C and 2F)

– Sperm mitochondrial fusion (Figure 5E and Figure 5—figure supplement 1C)

– FAD (Figure 4B-C)

The reduction of FAD biosynthesis pathway genes or *alh-6* in whole animal RNAi recapitulates all phenotypes. Germ cell specific RNAi of FAD biosynthesis pathway genes or *alh-6* recapitulated all sperm defects except sperm size (which seem to require some input from the somatic; discussion of this point is included). Riboflavin supplementation restores FAD level and suppresses all the sperm defects in the *alh-6* mutants (Figure 4E-G, Figure 5I).

4) Is the FAD level restored to WT in alh-6;prdh-1 double mutants? This is important to draw the conclusion that PRDH-1 continues to deplete FAD in the absence of alh-6 (subsection “FAD mediates sperm functionality and competitive fitness”).

Based on the previous review we have focused on *alh-6* and the changes in FAD as this is the novel aspect of our work.

As the reviewer is likely aware, P5C can be generated by ornithine and arginine metabolic pathways as well as proline catabolism. Although the *prdh-1* mutant would shed light on the latter there is evidence that homeostatic changes in these critical metabolic pathways occurs and we cannot rule out alterations to the urea cycle.

5) The subtitle "loss of cellular proline catabolism is not causal for sperm defects in alh-6 mutants" needs to be reconsidered. It appears contradictory to the title of the manuscript.

We have focused our study on the novel aspects relating to the loss of ALH-6, specifically the changes in FAD and cell autonomous roles in sperm.

6) The authors proposed that accumulation of P5C could account for ROS change that leads to sperm defects in alh-6 mutants. Experimental data should be provided on restoration of ROS levels by supplement with NAC and FAD.

Fixed and also refocused our study on FAD.

7) Are mitochondrial defects seen in alh-6 sperms diet-specific? The authors showed that partial inactivation of fzo-1 restored sperm activation, how about RNAi of eat-3? Does RNAi of fzo-1 and eat-3 change the ROS levels in alh-6 mutant sperms?

This is an excellent question that we apologize for not making clear in the first submission. The mitochondrial defect in *alh-6* sperm is not diet-specific and is manifested in animals fed either the OP50/*E. coli* B or HT115/K-12 diet (Figure 5E and Figure 5—figure supplement 1C).

*fzo-1* RNAi rescued both mitochondrial and activation defects in *alh-6* spermatids, while *eat-3* RNAi rescued just the mitochondrial defect but not the activation defect (Figure 5J-K, N-O). The connections between inner and outer mitochondrial membrane dynamics is continually emerging. The increase in *fzo-1* expression in *alh-6* mutant together with these results suggest that the defects in *alh-6* clearly engage the FZO-1 pathway.

Unfortunately, we could not measure ROS in sperm, however we have assessed the literature and found that *eat-3* mutant is more sensitive to paraquat than WT but not *fzo-1* mutant. This increased sensitivity in *eat-3* suggests a mitochondrial fusion independent role in ROS homeostasis. Furthermore, *eat-3* mutants have other defects that are enhanced by sod-2 mutation (Kanazawa et al., 2008).

8) The current model (Figure 6N) failed to define the relationships of P5C accumulation, FAD depletion, ROS change, mitochondrial dynamics and sperm quality. A better model needs to be suggested.

From all the new data and revisions we made to this manuscript we have come up with a model that clarifies the relationships between FAD, *alh-6*, and mitochondrial dynamics on sperm function. See Figure 7 for model.

[Editors’ note: what follows is the authors’ response to the second round of review.]

Reviewer #2:[…]1) It is interesting that loss of ALH-6, but not PRDH-1 which uses FAD as cofactor, caused reduction of FAD levels. What are the possible biochemical explanations? Can this be experimentally tested?

This is an important point. Our current and previous work has shown that the first step of proline catabolism (PRDH-1) is not reduced despite the loss of the second step in the pathway (ALH-6).

The changes in FAD are unlikely to be a result of the loss of any single enzyme. Instead, in the *alh-6* mutant background, the accumulation of P5C results in a stress response that includes the upregulation of many FAD binding enzymes as shown in Figure 3. It is this overall transcriptional signature that drives the phenotypes associated with *alh-6* loss. In other words, it is the downstream response to the loss of *alh-6*, not simply the loss of this enzyme, that is causal for this reduction in FAD. We have added additional text to explicitly state this hypothesis in the subsection “Transcriptional signatures define temporal phenotypes of *alh-6* mutant animals”.

2) The authors performed germline-specific RNAi to inactivate alh-6 and FAD synthesis genes, which induced sperm defects as in alh-6 mutant animals. In addition, expression of ALH-6 in the germline rescued the sperm defects. Thus, the conclusion is drawn that the effects of alh-6 loss on sperm function are cell autonomous. This seems not to be sufficient. Will overexpression of ALH-6 in somatic tissues, e.g., the intestine, rescue the sperm defect in alh-6 mutants?

The reviewer is correct that our model of cell autonomous function is based on our findings that the reduction of *alh-6* expression specifically in the germline recapitulating the *alh-6* mutant sperm defects when combined with expression of wildtype *alh-6* specifically in the germline can rescues the defects of *alh-6* mutants.

Our previous work has demonstrated that ALH-6 function in the soma is important for animal lifespan; specifically, loss of *alh-6* can impact muscle, intestine, and neuronal tissues. Although expression only in somatic tissues (i.e. germline restricted) is technically challenging (to our knowledge there doesn’t exist a “soma-specific” promoter), we address the reviewer’s comment by utilizing a germline RNAi deficient strain from Carolyn Philips and Gary Ruvkun’s labs, GR1948 – *mut-14(mg464);smut-1(tm1301)* V. This strain is competent for RNAi in somatic tissues, but not the germline (PMID: 24684932). In this strain, RNAi of *alh-6* only in the soma does not phenocopy the *alh-6* mutant sperm activity phenotypes. Thus loss of *alh-6* in somatic tissues does not drive sperm activation defects, but may contribute cell non-autonomously to cell size. This data can now be found in Figure 6—figure supplement 1A-B and further discussed in the subsection “*alh-6* and FAD are cell autonomous regulators of sperm function”

In conclusion, and in contrast to the germline specific experiments reported in the last submission, RNAi of *alh-6* only in the soma fails recapitulate the *alh-6* mutant sperm phenotype.

Reviewer #3:[…]1) Though Pronase (a mixture of proteases which cleave any surface protein on the cell) could be used to assess wild type sperm activation, Monensin is a better artificial sperm activation as it can make activated spermatozoa to crawl. At least, the authors need to check at which step mutant sperm failed to activate. The crossing data (in vivo sperm activation in Figure 1F) are inconsistent to the rest mutant sperm activation data.

We thank the reviewer for suggesting the use of Monensin. We have tested Monensin and similar to previous studies we find if to be a poorer activator of spermatids (Ward et al., 1983). Furthermore, it was also shown that spermatids activated by either treatment produces motile spermatozoa (Shakes and Ward, 1989). Nevertheless, we now include our tests of the *alh-6* mutant spermatids with Monensin treatment, which like our previous results with Pronase, display reduced activation compared to WT spermatids. This data can be found in Figure 2—figure supplement 1D and in the subsection “Defects in mitochondrial proline catabolism impact sperm quality”.

We now also include the assessment of intermediate stages of spermiogenesis. Our previous assessment was very strict, where the observance of “spikes” and rounded protrusions (Nelson and Ward, 1980; Shakes and Ward, 1989), were both considered not activated. Only the extension of a pseudopod was considered an activated state, noting that the treatment time of 30 min was sufficient for pseudopod formation in ~80% of WT spermatids. We have now further analyzed our data for *alh-6* mutants compared to WT and document the percent of intermediate activation states for both Pronase and Monensin treatments. Intriguingly, there does seem to be a difference in the type of stalled intermediate observed when either Pronase or Monensin is used. While Pronase treatment results in an increase in intermediates with spikes, Monensin treatment results in an increase of protrusions. These new data can be found in Figure 2—figure supplement 1B, C, and E and in the subsection “Defects in mitochondrial proline catabolism impact sperm quality”.

We believe that the data in Figure 1G (in vivo sperm activation) are consistent with sperm that are not completely abolished for sperm function, but are partially deficient in activation (note that 60% activate) and are simply smaller. The reduction in sperm activation and smaller size of *alh-6* male sperm may lead to disadvantages in competing against WT hermaphrodite sperm. This impairs, but does not abolish the ability to fertilize the next passing oocyte (increase in WT hermaphrodite self-sperm fertilized progeny in those mated to *alh-6* males) as shown in Figure 1F. However, when not forced to compete against hermaphrodite sperm, *alh-6* mutant sperm are competent for fertilization Figure 1—figure supplement 3A-B. We better explain these result in the subsection “*alh-6* fertility defects are sperm-specific”.

2) JC-1 staining (Figure 5A-D) was employed extensively in this study to show the mitochondrial fusion or not. This reviewer suggests confirming that with a GFP-tagged mitochondrial protein such as TOM20 to avoid any bias for imaging collection and analysis.

We thank the reviewer for this comment as it is an important point, which we now explain in text and include additional controls for specificity. JC-1 is the preferred staining method for spermatid mitochondria for several reasons, including:

1) JC-1 fluorescence (red) is specific for coupled mitochondria with a strong membrane potential (Smiley et al., 1991; Reers et al., 1995, Methods Enzymol.; Di Lisa et al., 1995, J. Physiol.; Cossarizza et al., 1996, Exp. Cell. Res.; Mathur et al., 2000, Cardiovasc. Res.).

2) We have tested several protein based mitochondria-targeted reporters (made by us and others in the field), but as the reviewer comments above, the transcriptional and translational quiescence of sperm limits the expression of these constructs thus prohibiting their use in quantitative assessments.

Nevertheless, to address the reviewers concern we now also show that treatment with the mitochondrial specific uncoupler Carbonyl cyanide m-chlorophenyl hydrazone (CCCP) abolishes the JC-1 fluorescence in the red channel, which confirms the mitochondria specificity of this dye. We better clarify in text that we use mitochondria with red JC-1 emission for assessment of connectivity of the healthiest mitochondria with strong membrane potential (subsection “Mitochondrial dynamics regulate spermatid function”, second paragraph). In addition, we include our data from earlier preliminary studies assessing mitochondrial connectivity with a different mitochondrial specific dye MitoTracker Red (Chen et al., 2003, J. Cell Bio.; Matsuda et al., 2010, J. Cell Bio.; Cottet-Rousselle et al., 2011, Cytometry A.). Although JC-1 is the preferred vital stain, MitoTracker staining results in a similar observation of increased connectivity in *alh-6* mutant sperm mitochondria as compared to wildtype counterpart. Taken together these data confirm the specificity of these dyes in mitochondrial assays as previously established by the field. These new data can be found in Figure 5—figure supplement 1A and C.

3) Figure 1—figure supplement 1D should use a better image instead of the current one.

We have replaced this image.